# Hepatic expression of GAA results in enhanced enzyme bioavailability in mice and non-human primates

Helena Costa-Verdera [1,2,3,7], Fanny Collaud [1,2,7], Christopher R. Riling[4,7], Pauline Sellier[1,2], Jayme M. L. Nordin[4], G. Michael Preston[4], Umut Cagin[1,2], Julien Fabregue[1,2], Simon Barral[1,2], Maryse Moya-Nilges[5], Jacomina Krijnse-Locker[5], Laetitia van Wittenberghe[1], Natalie Daniele[1], Bernard Gjata[1], Jeremie Cosette[1], Catalina Abad[6], Marcelo Simon-Sola[1,2], Severine Charles[1,2], Mathew Li[4], Marco Crosariol[4], Tom Antrilli[4], William J. Quinn III[4], David A. Gross[1,2], Olivier Boyer [6], Xavier M. Anguela[4], Sean M. Armour [4,8], Pasqualina Colella [1,2,8], Giuseppe Ronzitti [1,2,8] & Federico Mingozzi [1,2,3,4,8 ✉]

Pompe disease (PD) is a severe neuromuscular disorder caused by deficiency of the lysosomal enzyme acid alpha-glucosidase (GAA). PD is currently treated with enzyme replacement therapy (ERT) with intravenous infusions of recombinant human GAA (rhGAA). Although the introduction of ERT represents a breakthrough in the management of PD, the approach suffers from several shortcomings. Here, we developed a mouse model of PD to compare the efficacy of hepatic gene transfer with adeno-associated virus (AAV) vectors expressing secretable GAA with long-term ERT. Liver expression of GAA results in enhanced pharmacokinetics and uptake of the enzyme in peripheral tissues compared to ERT. Combination of gene transfer with pharmacological chaperones boosts GAA bioavailability, resulting in improved rescue of the PD phenotype. Scale-up of hepatic gene transfer to non-human primates also successfully results in enzyme secretion in blood and uptake in key target tissues, supporting the ongoing clinical translation of the approach.

[1] Genethon, 91000 Evry, France. [2] Université Paris-Saclay, Univ Evry, Inserm, Integrare research Unit UMR_S951, 91000 Evry, France. [3] Sorbonne University Paris and INSERM U974, 75013 Paris, France. [4] Spark Therapeutics, Philadelphia, PA 19104, USA. [5] Pasteur Institute, 75015 Paris, France. [6] Université de Rouen Normandie-IRIB, 76183 Rouen, France. [7] These authors contributed equally: Helena Costa-Verdera, Fanny Collaud, Christopher R. Riling. [8] These authors jointly supervised this work: Sean M. Armour, Pasqualina Colella, Giuseppe Ronzitti, Federico Mingozzi. ✉email: Federico.mingozzi@sparktx.com

Pompe disease (PD, acid maltase deficiency or glycogen storage disease type II; OMIM #232300) is a metabolic neuromuscular disorder caused by mutations in the gene encoding for the lysosomal enzyme acid alpha-glucosidase (GAA), which catalyzes the degradation of glycogen into glucose. Decreased GAA activity leads to pathological accumulation of glycogen, altering lysosomal function, and cell metabolism[1]. The severity of PD and life expectancy generally depend on the age of onset and the residual GAA enzyme activity[2]. The most severe classic infantile onset form of PD (IOPD; GAA activity <1%) presents with hypertrophic cardiomyopathy, muscle weakness, hypotonia and respiratory insufficiency leading to premature death if left untreated[2]. Late onset PD (LOPD, GAA activity up to 20–30%) presents at any time after 12 months of age with muscle weakness and respiratory dysfunction[2]. Several clinical studies have described the multi-systemic nature of PD, which includes central nervous system (CNS)[3,4] and vascular defects[5]. The current standard of care for PD consists of enzyme replacement therapy (ERT) based on bi-weekly intravenous infusions of recombinant human GAA (rhGAA, alglucosidase alfa, commercialized as lumizyme or myozyme)[1]. From the circulation, rhGAA (~110 kDa) is taken up by tissues via the cation-independent mannose-6-phosphate receptor (CI-M6PR) and trafficked to the lysosome[6], where it is proteolytically matured into its active forms (76 and 70 kDa)[7]. ERT with rhGAA has shown to resolve cardiomyopathy and improve the survival of IOPD subjects[8], as well as to improve and/or stabilize the disease progression in LOPD subjects[9,10]. Yet, ERT is not a cure for PD and the disease still remains lethal for the subset of IOPD subjects that develop neutralizing immune responses to rhGAA[9]. Moreover, in the long term, ERT-responsive IOPD patients often develop distal muscle weakness, respiratory dysfunction, and neurocognitive impairment, among others[11–13]. Disease progression with decline of locomotor ability, muscle and respiratory function has also been reported in ERT-treated LOPD subjects[14]. The short rhGAA half-life and its variable uptake in tissues, such as skeletal muscles, limit the efficacy of ERT[15,16]. The combination of pharmacological chaperones (PCs) with ERT has shown the potential to improve the rhGAA enzyme bioavailability in preclinical research[17] and clinical trials[18,19].

Gene therapy with adeno-associated virus (AAV) vectors provides the opportunity to turn the liver into a biofactory for the expression of therapeutic proteins, a concept demonstrated in preclinical studies[20,21] and currently explored in clinical trials (e.g., ClinicalTrials.gov NCT03533673, NCT04093349). We previously showed that AAV vector-mediated hepatic expression of a secretable form of GAA can correct the PD phenotype in mice[21,22]. However, a comparison of the efficacy of gene therapy to ERT in $Gaa^{-/-}$ mice is hampered by the development of anti-rhGAA antibodies and fatal hypersensitivity reactions to the infused enzyme[23]. To overcome this limitation, here we generated a colony of immunodeficient $Gaa^{-/-}$ mice. The PD mouse model allowed us to perform long-term, bi-weekly rhGAA infusions and to compare the pharmacokinetics, biodistribution, and therapeutic efficacy of ERT with AAV gene transfer.

Our work shows that hepatic gene therapy with AAV vectors has a superior therapeutic efficacy compared to ERT as it results in superior enzyme bioavailability. The steady-state expression in hepatocytes after gene transfer indeed resulted in therapeutically relevant enzyme biodistribution and glycogen clearance in ERT-refractory tissues, such as skeletal muscle and CNS. Enhanced enzyme bioavailability was also achieved through the combination of low dose AAV-GAA gene transfer with PCs. Based on these results, we developed an optimized vector based on a hepatotropic capsid and an improved secretable GAA transgene expression cassette. Hepatic gene transfer in non-human primates (NHPs) showed safe, dose-dependent secretion of GAA in plasma and enzyme uptake in peripheral tissues, supporting the translation of the approach to humans (ClinicalTrials.gov NCT04093349).

## Results

**Hepatic gene transfer drives superior rescue of the muscle phenotype compared to ERT in Pompe mice.** To generate a PD model which allows to compare ERT with AAV vector-mediated gene transfer, we bred $Cd4^{-/-}$ mice[24] with the mouse model of PD generated by Raben and colleagues[25] (Supplementary Fig. 1A). Analyses of wild-type ($Gaa^{+/+}Cd4^{-/-}$) and affected ($Gaa^{-/-}Cd4^{-/-}$) mice at 4 months of age confirmed the accumulation of glycogen in several tissues and the decrease in muscle strength in the colony (Supplementary Fig. 1B, C). As previously observed in PD mice[21], no significant alterations of respiratory function were detected in 4-month-old mice (Supplementary Fig. 1D). Due to the immunodeficient background, $Gaa^{-/-}Cd4^{-/-}$ mice were not expected to present the same proinflammatory features previously described in the original colony[21,22], although the $Cd4^{-/-}$ background is associated with a normal development of CD8+ T cells and an unaltered myeloid compartment[24].

$Gaa^{-/-}Cd4^{-/-}$ mice were treated with rhGAA at the dose of 20 mg/kg every 2 weeks, the standard of care regimen used in PD[9], for 4 months ($n = 9$), or with a single administration of AAV8 vector expressing secretable human GAA [AAV8-hAAT-sp7-Δ8-coGAA[21], hereby named AAV-GAA] at three different doses ($1 \times 10^{11}$, $5 \times 10^{11}$ or $2 \times 10^{12}$ vg/kg) at day 0 (Fig. 1A). Male animals were used in the study to minimize variability in liver transduction with AAV vectors across sexes[26]. A vector dose-dependent increase in circulating GAA activity was observed in AAV-treated animals (Fig. 1B). No GAA activity was detectable in ERT-treated animals seven days after ERT administration (Fig. 1B), in agreement with the clearance of rhGAA in plasma over time (Fig. 1C). Peak levels of GAA activity were reached 3 hours post ERT and matched the enzyme levels at plateau at the highest AAV vector dose ($2 \times 10^{12}$ vg/kg, Fig. 1C). In AAV-treated animals, a dose dependent vector genome copies increase was observed in liver (Fig. 1D). No animal in the study developed anti-hGAA IgG (Fig. 1E).

Because of the relatively low sensitivity of the GAA activity assay, we also used blood to perform a Western blot with an anti-human GAA antibody. At the end of the study, the GAA enzyme was barely detectable in ERT-treated animals, despite receiving a total of nine infusions (Fig. 1A, F, G). Circulating GAA was instead readily detected in all AAV-treated animals (Fig. 1F, G).

Next, we measured GAA activity in tissues at sacrifice. GAA activity levels were significantly elevated in heart and skeletal muscle of mice receiving the AAV-GAA vector at the dose of $2 \times 10^{12}$ vg/kg, while a slight increase was seen at $5 \times 10^{11}$ vg/kg (Fig. 2A). No differences in GAA activity were observed between ERT and the $1 \times 10^{11}$ vg/kg groups (Fig. 2A). The comparison of GAA activity across muscle groups confirmed the more efficient enzyme uptake in heart and diaphragm compared to quadriceps and triceps (Fig. 2B). Western blot analyses clearly showed higher amounts of lysosomal GAA protein in muscle tissues from AAV-treated mice compared to ERT-treated mice (Fig. 2C, D and Supplementary Fig. 2). Importantly, the detection of the 76 kDa lysosomal form of GAA in tissue lysates confirmed that the enzyme was correctly taken up by tissues and targeted to the lysosome[27].

The 4-month ERT treatment (Fig. 1A) normalized glycogen content in heart and partially in the diaphragm, while no effect was seen in skeletal muscle quadriceps and triceps (Fig. 2E). Conversely, an AAV vector dose-dependent correction of glycogen content was measured in all muscle tissues analyzed (Fig. 2E) and reflected enzyme content in tissues (Fig. 2A). Partial rescue of glycogen

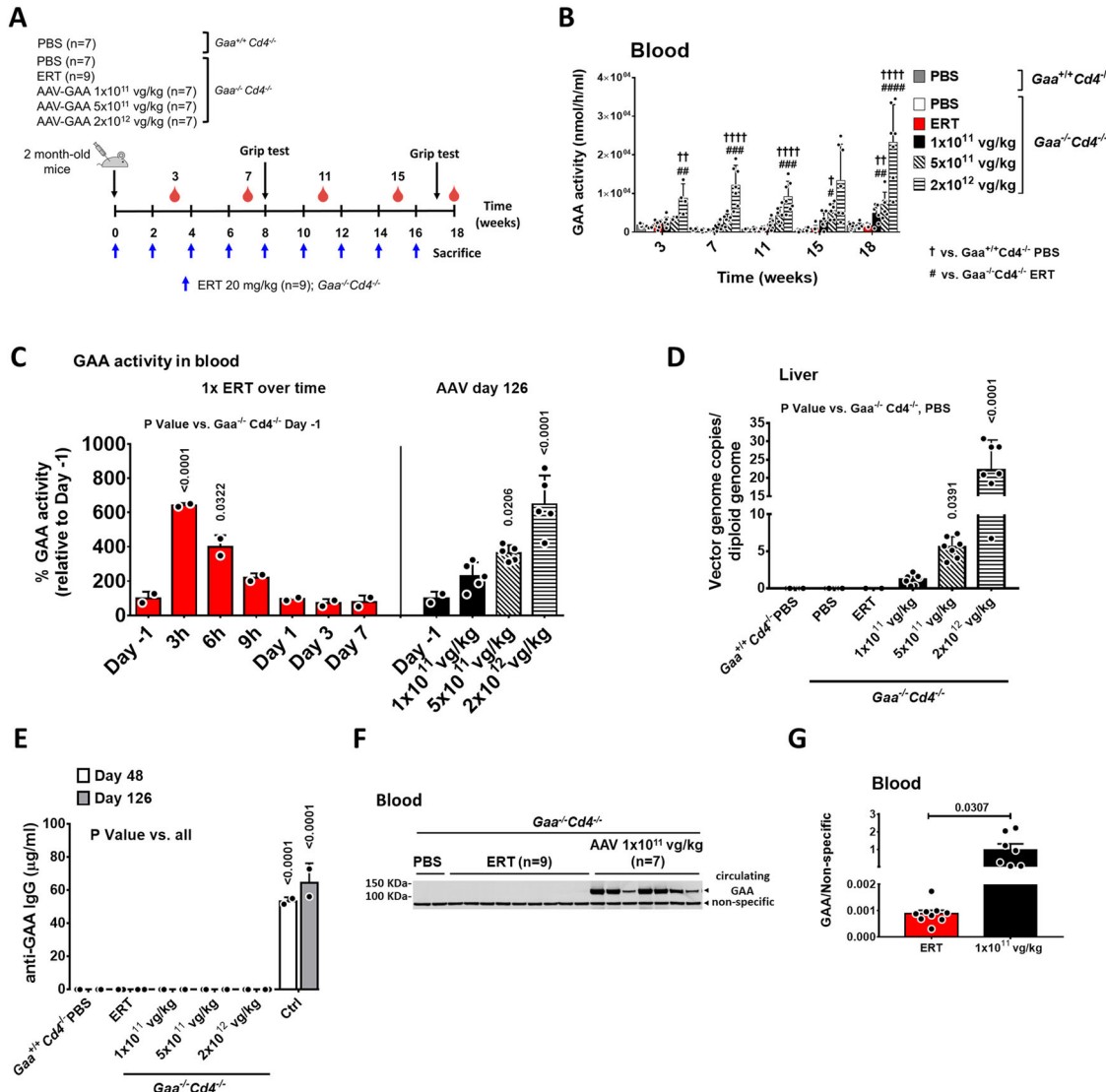

**Fig. 1 AAV-GAA gene transfer drives higher steady-state GAA levels in blood than enzyme replacement therapy (ERT). A** Experimental design. 2-month-old $Gaa^{-/-}Cd4^{-/-}$ mice received 9 bi-weekly injections of rhGAA at 20 mg/kg (blue arrows, $n = 9$) or a single treatment with AAV-GAA at day 0 ($n = 7$ per AAV dose). PBS-injected $Gaa^{+/+}Cd4^{-/-}$ ($n = 7$) and $Gaa^{-/-}Cd4^{-/-}$ ($n = 7$) mice were used as controls. Red symbols indicate the timing of blood collection in all cohorts. **B** Analysis of GAA activity in blood at the indicated time-points. **C** Comparison of the GAA activity in blood over time after ERT ($n = 2$) with the GAA activity measured at sacrifice in mice treated with AAV-GAA at the three doses indicated ($n = 5$ per AAV dose); Day −1, $n = 2$. **D** Analysis of vector genome copy number in liver at sacrifice. **E** Circulating anti human GAA IgG. $Gaa^{-/-}Cd4^{+/+}$ mice injected with an AAV vector expressing the GAA transgene under the control of the CAG promoter were used as immunization controls (Ctrl, $n = 2$). **F** Western blot analysis of circulating GAA protein in $Gaa^{-/-}Cd4^{-/-}$ mice treated with ERT or $1 \times 10^{11}$ vg/kg AAV-GAA at 18 weeks. **G** Quantification of GAA protein bands in **F**. Data shown as mean ± SD. Statistical analysis: **B**, **E** Two-way ANOVA with Tukey post-hoc; **C**, **D** One-way ANOVA with Tukey post-hoc; **G** t-test; † and # $p < 0.05$; †† and ## $p < 0.01$; ### and #### $p < 0.001$; †††† and #### $p < 0.0001$. Exact $p$ values for **B** are provided in the Source Data file. Number of animals per group in **B**, **D–G** as indicated in **A**.

accumulation was seen in quadriceps and triceps at the lowest vector dose of $1 \times 10^{11}$ vg/kg, while normalization was seen in the mid-vector and high-vector dose cohorts ($5 \times 10^{11}$ and $2 \times 10^{12}$ vg/kg, respectively) (Fig. 2E). Heart glycogen clearance in all treated groups (Fig. 2E) correlated with the normalization of the heart to body weight ratio (Fig. 3A). Two months after treatment, skeletal muscle function was already improved in the mid-dose and high-dose AAV-GAA cohorts compared to PBS-treated $Gaa^{-/-}Cd4^{-/-}$ animals (Fig. 3B). Four months after vector injection, all AAV-treated groups showed significantly improved muscle strength compared to untreated controls (Fig. 3C). Muscle strength in ERT-treated mice remained undistinguishable from untreated $Gaa^{-/-}Cd4^{-/-}$ mice (Fig. 3B, C) and was significantly lower than in mice treated with the

highest AAV vector dose (Fig. 3C). Average muscle fiber size and fiber size distribution was then measured to evaluate pathological muscle degeneration/renewal processes[17]. Laminin-stained quadriceps sections showed a normalization of fiber size and fiber size distribution in mice treated with AAV-GAA vector at a dose of $5 \times 10^{11}$ and $2 \times 10^{12}$ vg/kg, with a trend for improvement at the lowest vector dose (Fig. 3D–F). No significant correction observed in ERT-treated animals (Fig. 3D–F).

Overall, these results show that, in Pompe mice over an 18-week follow-up, hepatic expression of secretable GAA provides a superior correction of the skeletal muscle phenotype, despite the high peak enzyme activity levels observed with 20 mg/kg rhGAA infusions (Fig. 1C).

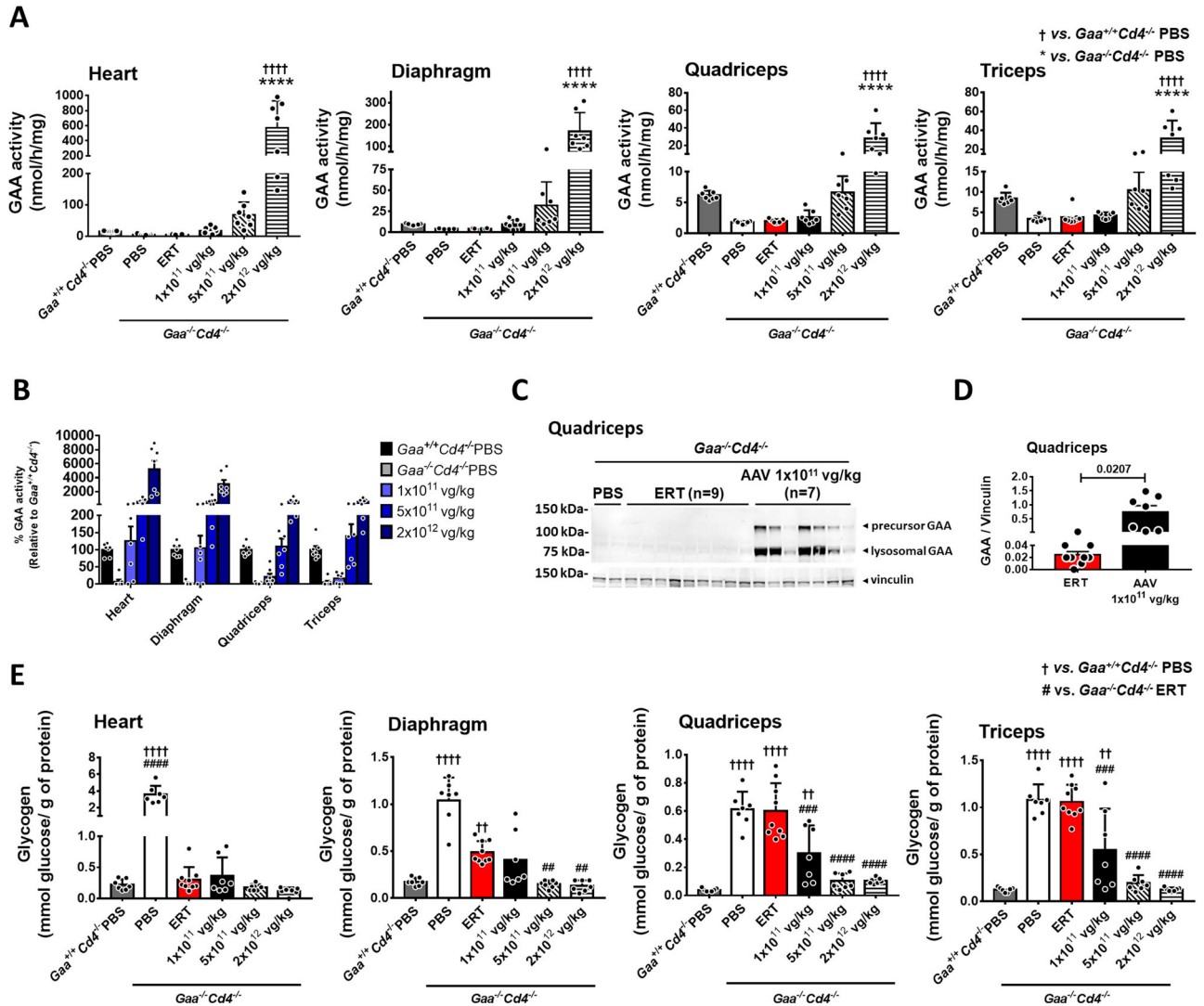

**Fig. 2 AAV-GAA gene transfer results in efficient GAA uptake and glycogen clearance in muscle. A** Analysis of GAA activity in muscles at sacrifice. **B** Comparison of GAA activity in different muscle groups. **C** Comparison of lysosomal GAA at sacrifice in quadriceps of mice treated with enzyme replacement therapy (ERT) vs. AAV-GAA gene transfer at $1 \times 10^{11}$ vg/kg. **D** Quantification of the lysosomal GAA bands depicted in **C**. **E** Analysis of glycogen content in muscles at sacrifice. PBS-injected $Gaa^{+/+}Cd4^{-/-}$ and $Gaa^{-/-}Cd4^{-/-}$ mice were used as controls. Data shown as average ± SD. Statistical analysis: **A**, **E** One-way ANOVA with Tukey post-hoc test; **D** t-test. †† and ## $p < 0.01$; ### $p < 0.001$; ****, ††††, and #### $p < 0.0001$. Exact $p$ values for **A** and **E** are provided in the Source Data file. **A**, **B**, **D**, **E**: ERT, $n = 9$; AAV-GAA, $n = 7$ per AAV dose; $Gaa^{+/+}Cd4^{-/-}$ PBS, $n = 7$, and $Gaa^{-/-}Cd4^{-/-}$ PBS, $n = 7$.

**Hepatic AAV gene transfer rescues glycogen accumulation in brain and spinal cord of Pompe mice.** Glycogen accumulation in the central nervous system (CNS) is emerging as a feature contributing to PD pathology in humans[15,28]. We observed a significant increase in GAA activity in brain and spinal cord of $Gaa^{-/-}Cd4^{-/-}$ mice treated with AAV-GAA at the dose of $2 \times 10^{12}$ vg/kg (Fig. 4A, B). Western blot analyses in brain and spinal cord showed the uptake and lysosomal targeting of GAA (Fig. 4C–F). Notably the amount of lysosomal GAA protein was significantly higher in animals treated with AAV-GAA than those treated with the ERT regimen, even at the lowest AAV vector dose tested (Fig. 4C–F).

The analysis of glycogen content in brain (Fig. 4G) showed a partial but significant dose-dependent decrease of glycogen accumulation in animals treated with AAV-GAA compared to PBS-treated $Gaa^{-/-}Cd4^{-/-}$ mice. A marked, dose-dependent decrease of glycogen accumulation was also observed in the spinal cord of AAV-treated animals (Fig. 4H). Notably, in this tissue, complete

correction of glycogen content was achieved in the AAV-GAA cohorts (Fig. 4H). Differently from AAV-GAA gene transfer, the 4-month ERT regimen did not result in a significant amelioration of glycogen accumulation in brain and spinal cord (Fig. 4G, H). These findings are consistent with previous reports showing that the intravenous administration of rhGAA does not correct the CNS pathology in mice[29].

Together, these results show that, differently from hepatic AAV gene transfer with secretable GAA, ERT is inefficient in clearing glycogen in the CNS of PD mice when given at the dose of 20 mg/kg every 2 weeks.

**Hepatic AAV gene transfer provides better rescue of lysosomal ultrastructure, autophagy, and mitophagy compared to ERT in mice.** PD is associated with significant autophagy[30,31] and mitophagy[32] impairment and alterations of lysosome ultrastructure[30]. We evaluated muscle ultrastructure and lysosome length in *tibialis anterior* collected at week 18. Enlarged

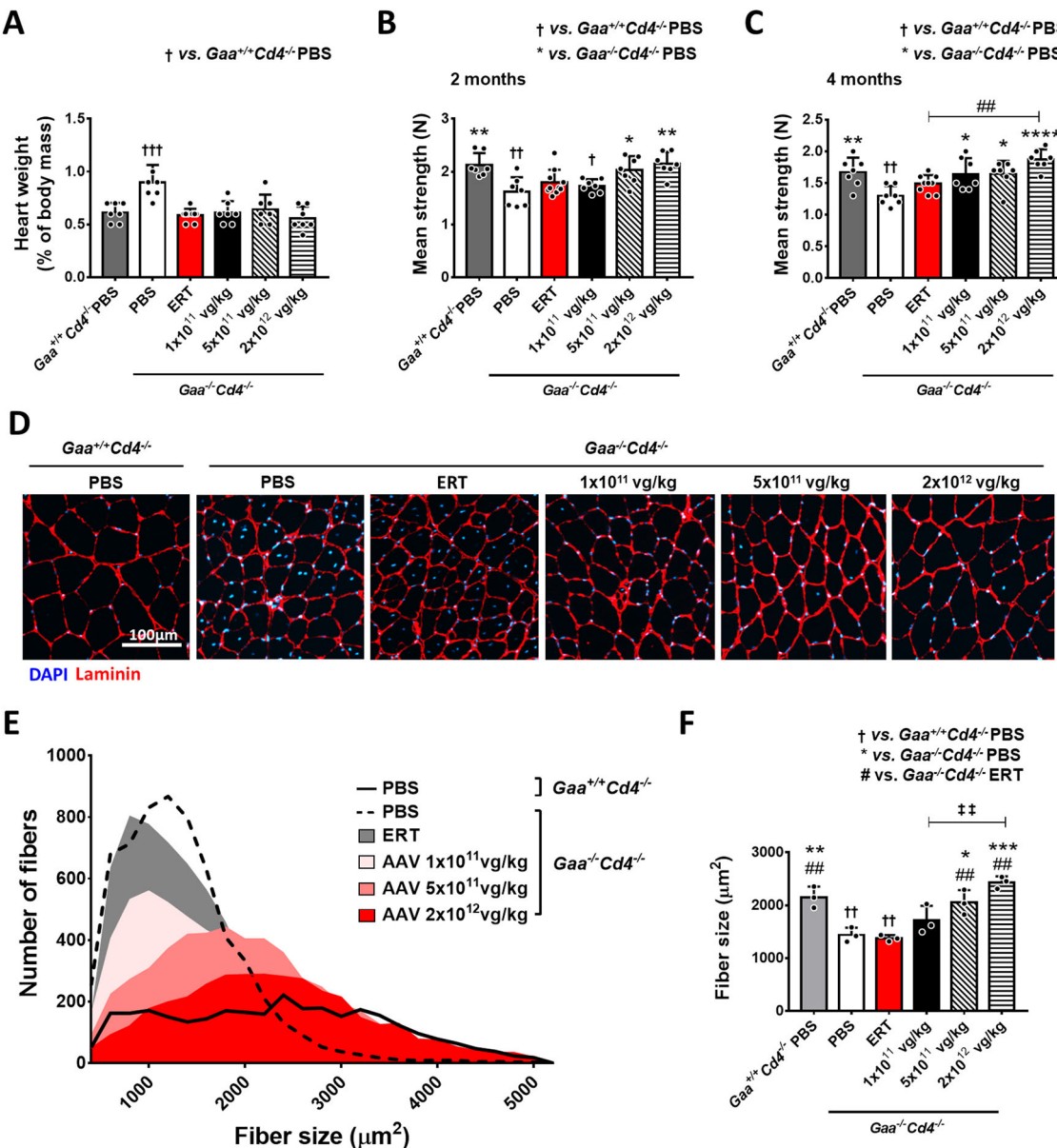

**Fig. 3 AAV-GAA gene transfer provides better rescue of skeletal muscle function and fiber size compared to enzyme replacement therapy (ERT).**
**A** Heart weight expressed as percentage of total body weight. **B**, **C** Analysis of grip strength at 2 and 4 months after treatment (Fig. 1A), respectively. **D** Representative immunofluorescence staining of muscle fibers from quadriceps sections with anti-laminin (red) and DAPI (blue), the results shown were reproducible across all animals in the corresponding treatment groups. **E** Representation of the muscle fiber size distribution across treatment groups at sacrifice. **F** Quantification of the mean fiber size area ($n = 3$ mice per group). Data shown as average ± SD. Statistical analysis: **A**–**C**, **F** One-way ANOVA with Tukey post-hoc test. *, † $p < 0.05$; **, ‡‡, ## and †† $p < 0.01$; ***, ††† $p < 0.001$; ****$p < 0.0001$. Exact $p$ values for **A**–**C**, **F** are provided in the Source Data file. **A**, **B**, **C**, **F**: ERT, $n = 9$; AAV-GAA, $n = 7$ per AAV dose; $Gaa^{+/+}Cd4^{-/-}$ PBS, $n = 7$, and $Gaa^{-/-}Cd4^{-/-}$ PBS, $n = 7$.

lysosomes, typical of PD muscle[30], were observed in PBS-treated $Gaa^{-/-}Cd4^{-/-}$ mice (Fig. 5A, B). Lysosomal length was significantly reduced in animals treated with the lowest AAV vector dose of $1 \times 10^{11}$ vg/kg (Fig. 5A, B), while complete normalization was observed at the mid and high vector doses ($5 \times 10^{11}$ vg/kg and $2 \times 10^{12}$ vg/kg) (Fig. 5A, B). No correction of lysosomal length was found in animals treated with ERT (Fig. 5A, B).

We next evaluated autophagy and mitophagy in skeletal muscle by measuring the content of p62[33] and Parkin[32] proteins, respectively. ERT administration showed a minimal decrease of p62 accumulation, while a significant p62 reduction was observed in $Gaa^{-/-}Cd4^{-/-}$ mice treated with AAV-GAA (Fig. 5C, D). Parkin amounts were normalized in $Gaa^{-/-}Cd4^{-/-}$ mice treated

with AAV-GAA at the mid and high vector doses (Fig. 5C, E). No significant differences were observed in animals treated with ERT or low dose AAV-GAA (Fig. 5C, E). Based on the evidence of mitophagy defects, we next evaluated the content of reactive oxygen species (ROS) in skeletal muscle by measuring the generation of mitochondrial free radicals by electron paramagnetic resonance (EPR)[34] (Fig. 5F). A significant increase in free radicals was measured in $Gaa^{-/-}Cd4^{-/-}$ mice compared to $Gaa^{+/+}Cd4^{-/-}$ mice (Fig. 5F), consistent with an accumulation of dysfunctional mitochondria. Differently from ERT treatment, AAV-GAA reduced the accumulation of free radicals in a dose-dependent manner with significant differences at the mid and high vector doses (Fig. 5F).

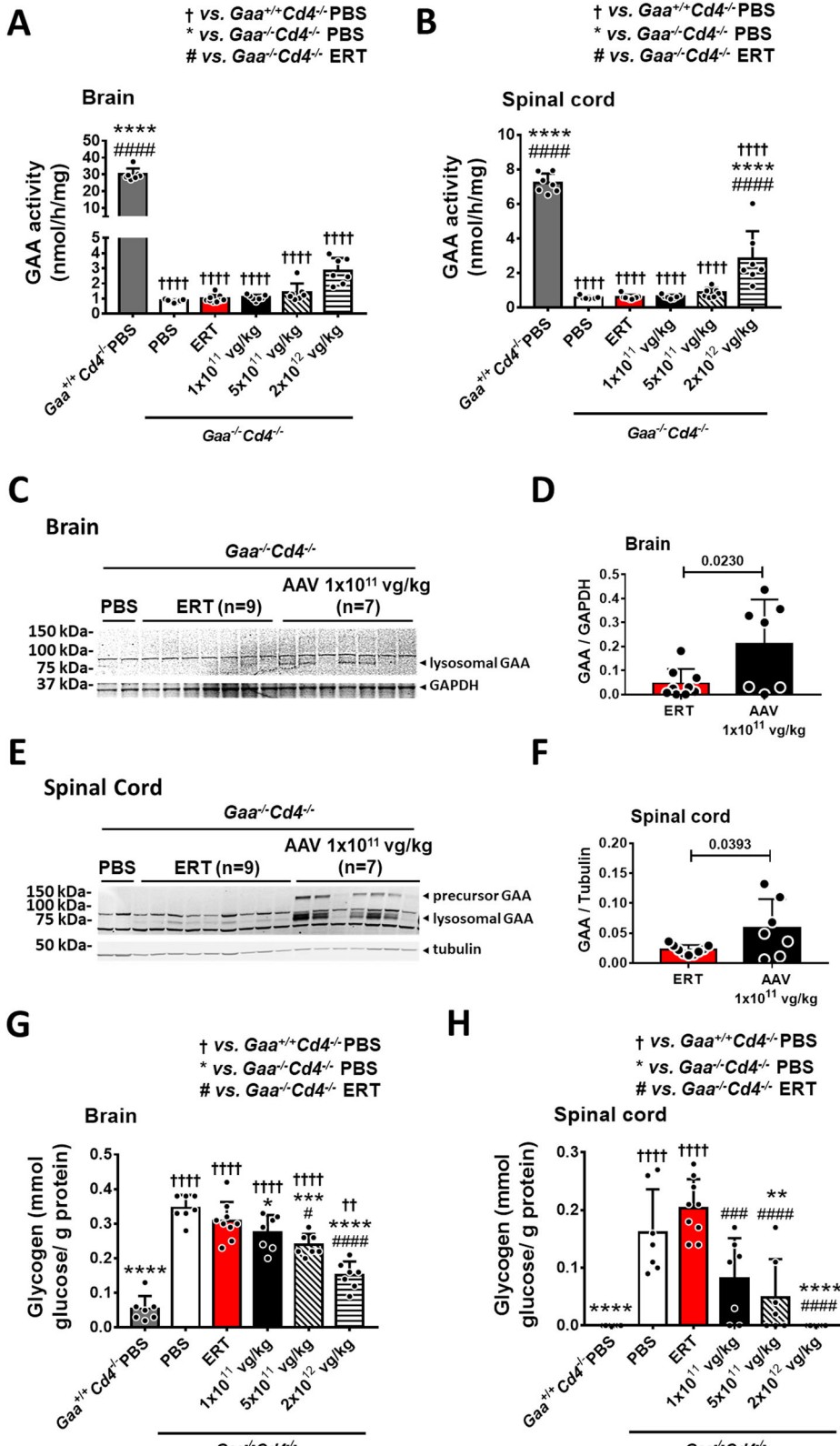

**Fig. 4 AAV-GAA gene transfer results in higher central nervous system uptake of GAA and glycogen clearance compared to enzyme replacement therapy (ERT). A** GAA activity in brain and **B** spinal cord at sacrifice. **C–F** Analysis of lysosomal GAA enzyme at sacrifice in brain (**C**, **D**) and spinal cord (**E**, **F**) of mice treated with ERT or AAV-GAA gene transfer at $1 \times 10^{11}$ vg/kg. **D**, **F** Quantification of the lysosomal GAA bands in **C** and **E**. **G**, **H** Analysis of glycogen content in brain (**G**) and spinal cord (**H**). Data shown as average ± SD. Statistical analysis: **A**, **B**, **G**, **H** One-way ANOVA with Tukey post-hoc test; **D**, **F** t-test. *, # and # $p < 0.05$; ** and †† $p < 0.01$; *** and ### $p < 0.001$; ****, †††† and #### $p < 0.0001$. Exact $p$ values for **A**, **B**, **G**, **H** are provided in the Source Data file. **A**, **B**, **D**, **F–H**: ERT, $n = 9$; AAV-GAA, $n = 7$ per AAV dose; Gaa$^{+/+}$Cd4$^{-/-}$ PBS, $n = 7$, and Gaa$^{-/-}$Cd4$^{-/-}$ PBS, $n = 7$.

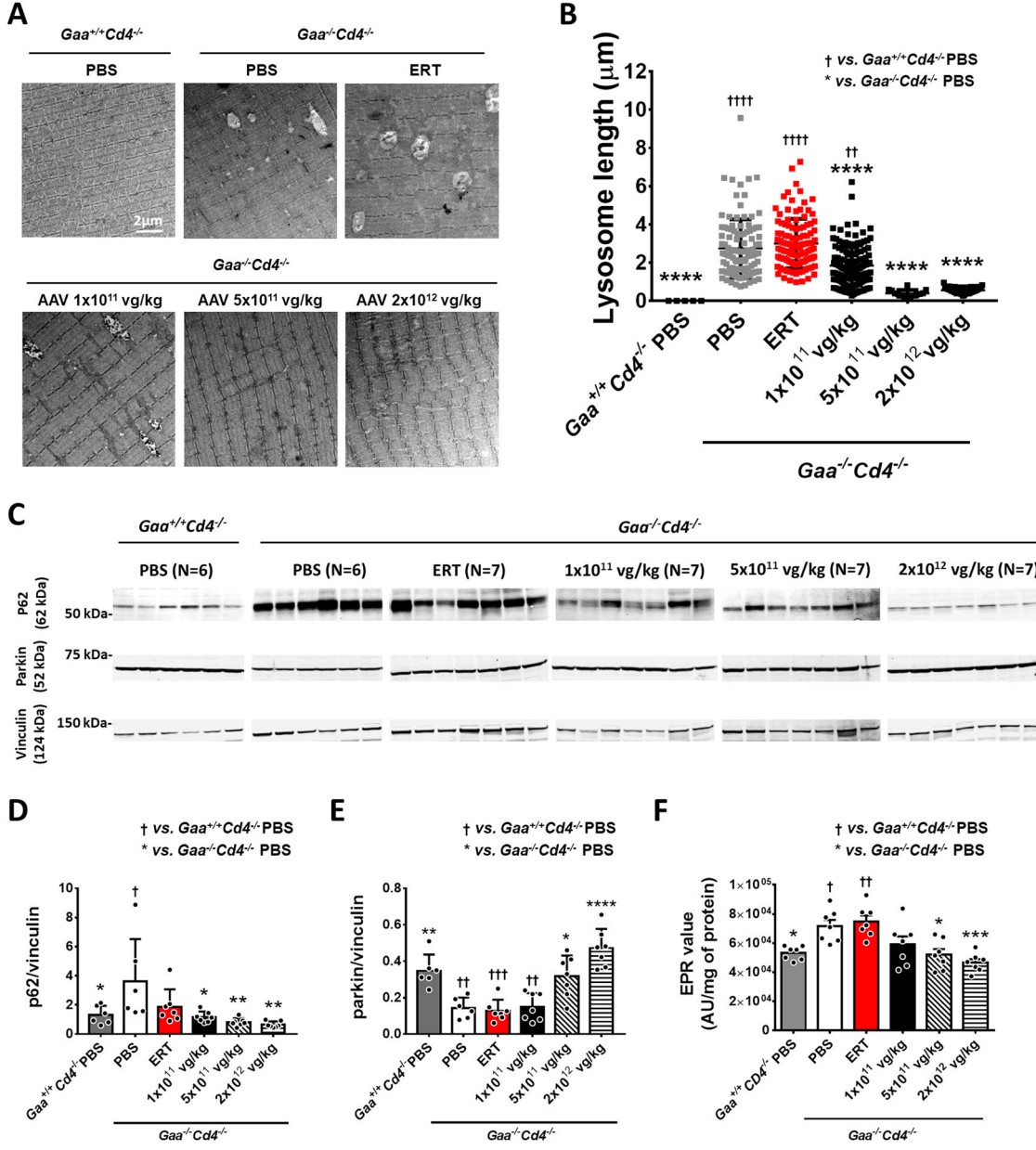

**Fig. 5 AAV-GAA gene transfer provides better correction of lysosomal enlargement and mitochondrial function compared to enzyme replacement therapy (ERT). A** Representative electron microscopy scans of *tibialis anterior* sections. **B** Quantification of lysosomal length from electron microscopy images (n = 2 per treatment group; number of lysosomes counted and used for statistical analysis: *Gaa*+/+*Cd4*−/− PBS, n = 4; *Gaa*−/−*Cd4*−/− PBS, n = 108; ERT, n = 113; AAV-GAA 1 × 10^11 vg/kg, n = 136; AAV-GAA 5 × 10^11 vg/kg, n = 14; AAV-GAA 2 × 10^12 vg/kg, n = 37). **C** Western blot analysis of p62 and Parkin amounts in quadriceps. Quantification of p62 (**D**) and Parkin (**E**) protein bands. **F** Analysis of reactive oxygen species (ROS) in *tibialis anterior*, expressed as arbitrary units (AU) per mg of protein (n = 7 per treatment group). Data shown as average ± SD. Statistical analysis: One-way ANOVA with Tukey post-hoc test. * and † $p < 0.05$; ** and †† $p < 0.01$; *** and ††† $p < 0.001$; **** and †††† $p < 0.0001$. Exact p values for **B**, **D**-**F** are provided in the Source Data file. **D**, **E**: ERT, n = 7; AAV-GAA, n = 7 per AAV dose; *Gaa*+/+*Cd4*−/− PBS, n = 6, and *Gaa*−/−*Cd4*−/− PBS, n = 6.

Together, these results indicate that, in skeletal muscle from Pompe mice, hepatic AAV-GAA gene transfer results in a superior rescue of lysosomal enlargement, autophagy block, and mitochondrial dysfunction compared to a 4-month ERT regimen.

**Pharmacokinetics of GAA in blood of Pompe mice following ERT or AAV-GAA gene transfer**. To provide mechanistic insights into the higher therapeutic efficacy observed with AAV-GAA gene transfer compared to repeated ERT infusions, we evaluated and compared the pharmacokinetics of GAA in the

context of the two therapeutic modalities. We measured the GAA activity and antigen over time after one rhGAA infusion or hepatic gene therapy with AAV-GAA at the lowest vector dose tested in this study. After a single intravenous infusion of rhGAA at 20 mg/kg (ERT cohort, n = 5), blood was collected at 3, 6, 9 h and 1, 3, 7, 14, and 21 days (Fig. 6A). In parallel, a separate cohort of mice received the AAV-GAA vector (n = 5) and was bled at day 3, 7, 14, 68, and 126 days (Fig. 6A). We also included in the analysis blood samples from mice treated with AAV-GAA at 5 × 10^11 vg/kg and 2 × 10^12 vg/kg from the previous study (Fig. 1A), in order to compare the levels of GAA in circulation at plateau among groups at 126 days after vector administration.

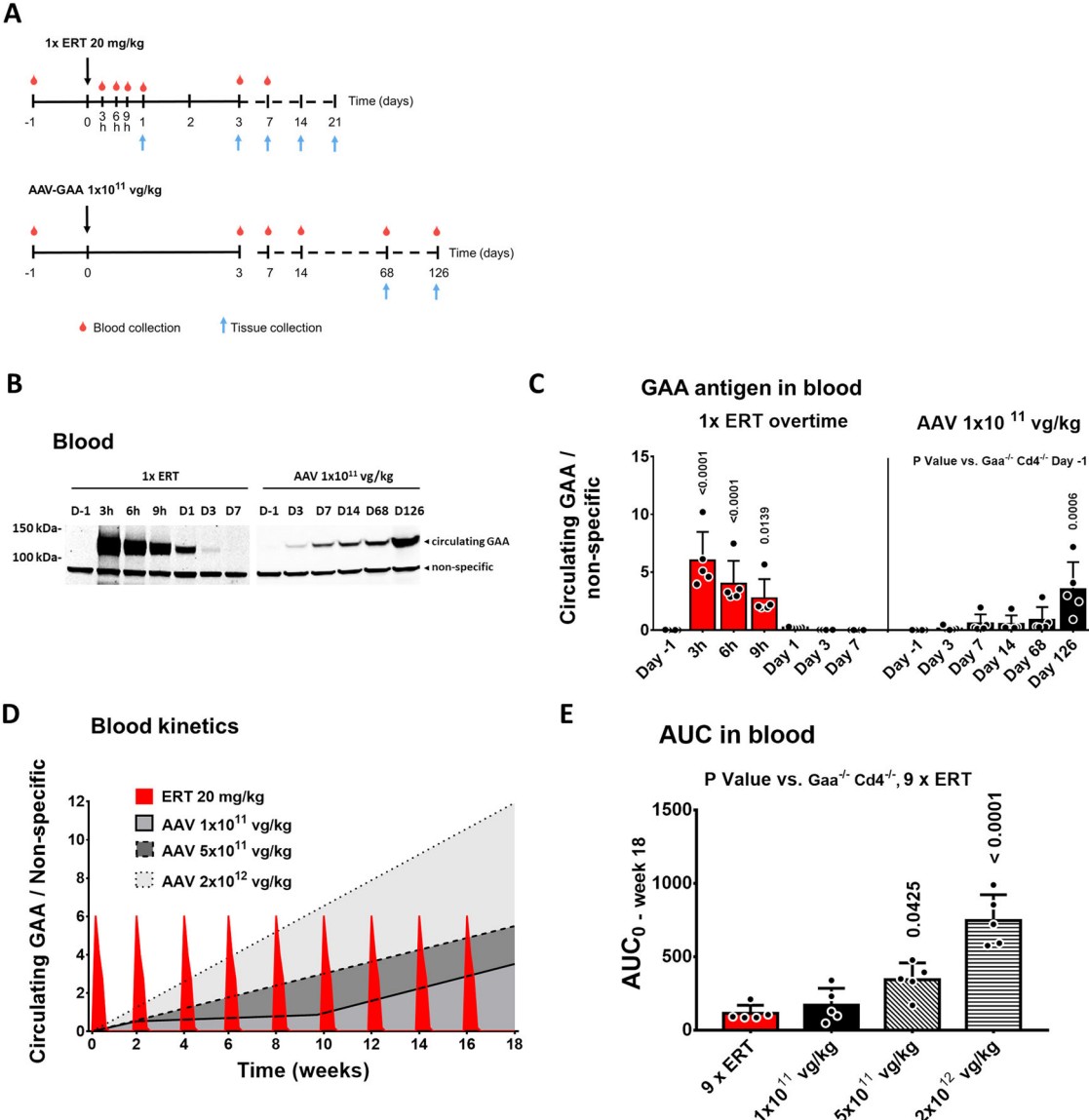

**Fig. 6 GAA kinetics in blood after enzyme replacement therapy (ERT) or AAV-mediated liver gene transfer. A** Experimental design. Two-month-old $Gaa^{-/-}Cd4^{-/-}$ male mice received a single injection of ERT at 20 mg/kg or a single injection of AAV-GAA at $1 \times 10^{11}$ vg/kg. **B** Example of Western blot analysis of circulating GAA ($n = 1$ per time point). **C** Quantification of circulating GAA enzyme in the corresponding Western blots (Fig. 6B and Supplementary Fig. 3C, D). **D** Kinetics of circulating GAA, measured by quantification of GAA enzyme in blood over time by Western blot. **E** Analysis of the predicted area under the curve (AUC) between time 0 (day −1) and week 18 for the different treatment groups. Data shown as average ± SD. Statistical analysis: One-way ANOVA. **C–E**: $n = 5$ mice per time point.

In ERT-treated mice, GAA protein measured by Western blot in blood peaked 3 hours post-infusion and became undetectable after three days (Fig. 6B, C and Supplementary Fig. 3A–C). Side-by-side GAA level measurements in blood showed that in animals treated with AAV-GAA at $1 \times 10^{11}$ vg/kg, the GAA protein in plasma gradually raised for the duration of the study, although at levels that were lower than those at peak ERT (Fig. 6B, C and Supplementary Fig. 3C, D).

To model the kinetics of the human GAA enzyme in the circulation over the 4-month study following either ERT or AAV-GAA administration, we plotted the amount of enzyme detected by Western blot (Fig. 6B, C and Supplementary Fig. 3) vs. the time points of measurement (Fig. 6D). To calculate the area under the curve (AUC) in the ERT study (Fig. 6D, E), we multiplied the data generated with a single rhGAA infusion (Fig. 6C) for the total number of infusions performed in the 4-month study (nine

infusions, Fig. 1A). For AAV gene therapy, we used the data generated at the lowest dose (Fig. 6B, C and Supplementary Fig. 3D) and assumed a linear increase for the $5 \times 10^{11}$ and $2 \times 10^{12}$ vg/kg dose-cohorts (Fig. 6D, E). Under these assumptions AUC was calculated from day 0 to week 18 (Fig. 6D, E). The AUC of GAA in the circulation was similar in ERT-treated mice and those receiving the lowest AAV-GAA dose ($1 \times 10^{11}$ vg/kg, Fig. 6D, E). The AUC was instead significantly higher in gene therapy-treated animals vs. the ERT cohort at the intermediate and high vector doses, $5 \times 10^{11}$ and $2 \times 10^{12}$ vg/kg, respectively (Fig. 6D, E).

Overall, these results suggest that hepatic gene transfer with secretable GAA at the vector dose of $1 \times 10^{11}$ vg/kg results in similar total exposure to circulating GAA enzyme to ERT over a follow up of 18 weeks. Superior exposure to circulating GAA is achieved at AAV vector doses equal or higher than $5 \times 10^{11}$ vg/kg.

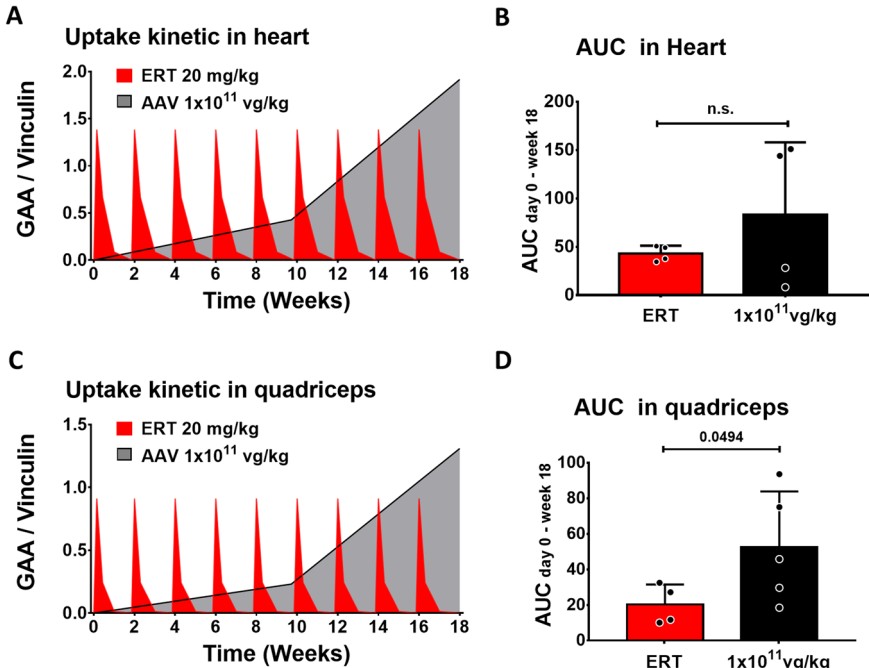

**Fig. 7 Hepatic gene transfer provides superior bioavailability of GAA in refractory tissues compared to enzyme replacement therapy (ERT). A** Representation of GAA kinetics in heart measured by quantification of lysosomal GAA enzyme over time by Western blot. **B** Analysis of the area under the curve (AUC) between time 0 (day −1) and week 18 in heart (ERT, $n = 4$; AAV-GAA, $n = 4$). **C** Representation of GAA kinetics in quadriceps (ERT, $n = 4$; AAV-GAA, $n = 5$). **D** Analysis of the area under the curve (AUC) in quadriceps. Data shown as average ± SD. Statistical analysis: $t$-test; n.s. non-significant.

**AAV-mediated gene transfer drives higher GAA bioavailability than ERT in skeletal muscle of PD mice.** To evaluate the kinetics of exposure of muscle to GAA, we measured lysosomal levels of the enzyme. Muscle samples were collected 1, 3, 7, 14, or 21 days after ERT or 68 and 126 days after AAV vector administration, when hepatic expression of GAA was well established (Fig. 6A). In the heart, a tissue known to be permissive to ERT[35], lysosomal GAA was detected up to 7 days post-ERT, but not thereafter (Supplementary Fig. 4A, B). In quadriceps, a tissue generally refractory to ERT[35], lysosomal GAA was detected up to 3 days post-rhGAA infusion (Supplementary Fig. 4C,D). In animals treated with AAV-GAA at $1 \times 10^{11}$ vg/kg, lysosomal GAA was detected at increasing amounts over time (Supplementary Fig. 4A, D), consistent with previous reports[21,22]. As expected, the administration of AAV-GAA doses of $5 \times 10^{11}$ and $2 \times 10^{12}$ vg/kg also resulted in a dose-dependent enzyme accumulation in the quadriceps (Supplementary Fig. 4E, F). The determination of the AUC in heart (Fig. 7A, B) showed that the exposure to GAA was not significantly different in animals treated with ERT and AAV gene therapy at the lowest vector dose tested. This observation was consistent with the glycogen clearance data in this tissue (Fig. 2E). Importantly, in quadriceps, where the rescue of glycogen accumulation was superior in AAV-treated vs. ERT-treated mice (Fig. 2E), a significant, 2.5-fold higher AUC was measured in the context of AAV-GAA gene transfer compared to ERT (Fig. 7C, D).

These results demonstrate that the superior therapeutic efficacy observed in skeletal muscle with gene therapy compared to ERT may be explained by the higher exposure to the therapeutic enzyme achieved through steady-state expression of secretable GAA in hepatocytes.

**Pharmacological chaperones enhance the efficacy of AAV-GAA gene transfer in PD mice.** To further increase the bioavailability of the GAA enzyme, we investigated the potential synergistic effect of combining pharmacological chaperones (PCs) with AAV-GAA hepatic gene transfer. We evaluated the combined action of 1-Deoxynojirimycin (DNJ, duvoglustat) and ambroxol (ABX), previously reported to improve GAA activity either in vitro[36] or in the context of ERT with rhGAA[18,19,37,38].

We first confirmed the enhanced bioavailability of GAA after administration of DNJ and ABX in ERT-treated $Gaa^{-/-}Cd4^{-/-}$ mice (Supplementary Fig. 5A). As expected, combined PC administration resulted in higher GAA activity and antigen in blood compared to ERT alone (Supplementary Fig. 5B, C).

Next, we evaluated the effect of the effect of PC treatment on AAV-GAA gene transfer. Mice received $1 \times 10^{11}$ vg/kg of AAV-GAA with or without the PCs (Fig. 8A). Blood analyses over time showed an increase in circulating GAA activity (Fig. 8B) and antigen (Fig. 8C, D) in PCs-treated animals. VGCN and GAA activity in liver was similar in animals treated with AAV-GAA regardless of the treatment (Supplementary Fig. 5D, E). Across several muscle groups, GAA activity was significantly higher in mice treated with PCs (Fig. 8E). Consistently, PC treatment resulted in better glycogen clearance in most muscles (Fig. 8F), with complete rescue of cardiomegaly observed only in animals that received PCs (Fig. 8G). Muscle strength was partially but significantly improved in all AAV-treated animals (Fig. 8H), regardless of PC administration.

Analysis of GAA activity in the CNS showed higher enzyme activity in the spinal cord of mice treated with PCs (Supplementary Fig. 5F). Despite this, no significant reduction of pathological glycogen accumulation was seen (Supplementary Fig. 5G), likely due to the short duration of the study[21].

Together, these results further support the hypothesis that stable circulating levels of GAA increase the enzyme bioavailability in refractory tissues. They also show the improved efficacy of a therapeutic strategy based on the combination of AAV gene transfer with a pharmacological treatment enhancing GAA enzyme stability and biodistribution.

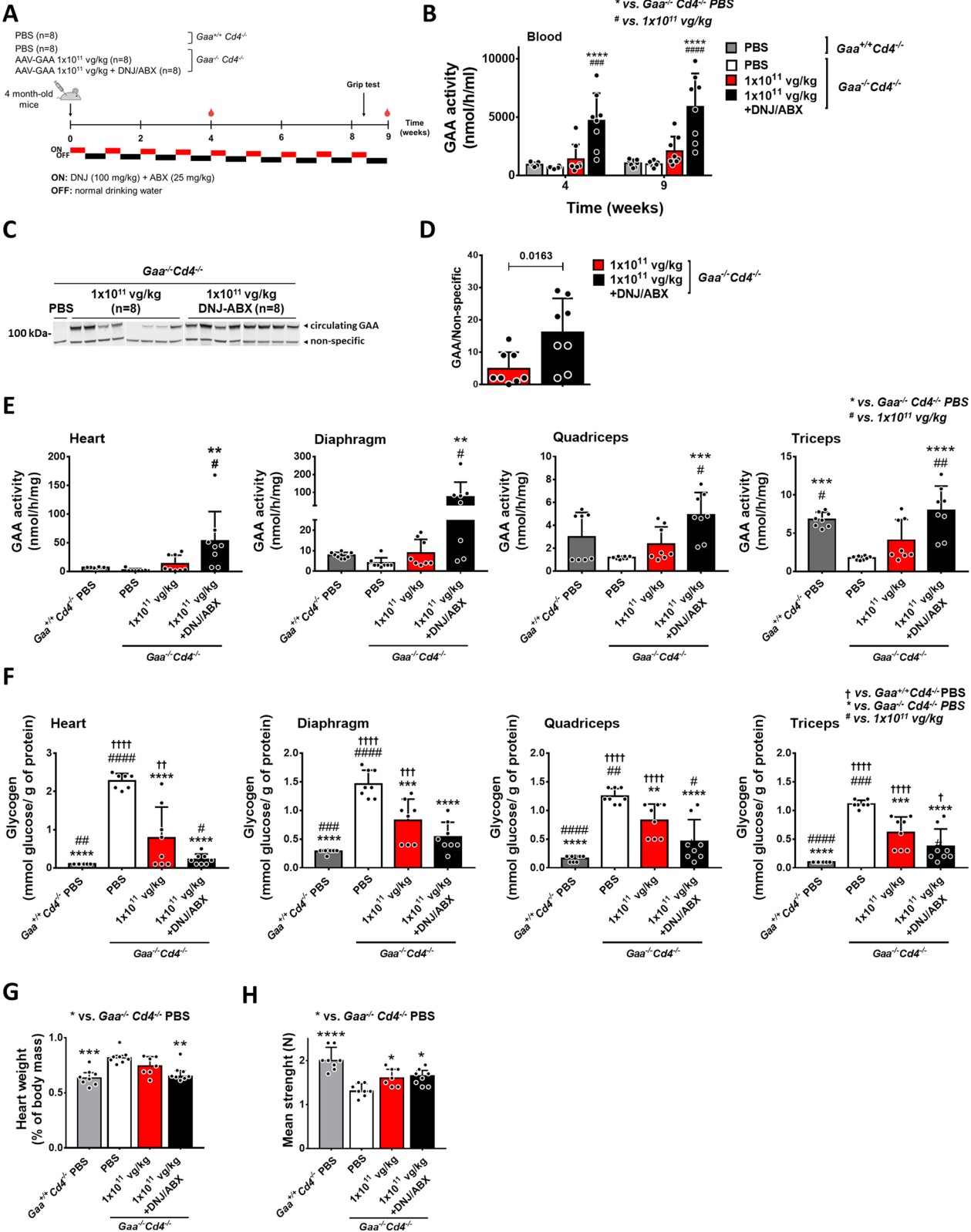

**Translational optimization and scale up to non-human primates (NHPs) demonstrate tissue uptake of hepatocyte expressed GAA.** In an effort toward clinical translation, we codon-optimized the GAA cDNA sequence initially tested in Pompe mice. The transgene expression cassette was then packaged into an AAV-Spark100 capsid, which has demonstrated hepatocyte tropism in humans[39]. The optimized AAV vector was

named SPK-3006. SPK-3006 was initially tested in wild-type C57BL/6 mice at the dose of $2 \times 10^{12}$ vg/kg, showing higher amounts of circulating GAA enzyme compared to the parental AAV-GAA vector (Fig. 9A and Supplementary Fig. 6A).

The SPK-3006 vector was then administered intravenously to healthy male Rhesus macaques (*Macaca mulatta*) at three ascending doses of $2 \times 10^{12}$, $6 \times 10^{12}$, and $2 \times 10^{13}$ vg/kg.

**Fig. 8 Pharmacological chaperones enhance gene therapy efficacy. A** Experimental design. 4-month-old $Gaa^{-/-}Cd4^{-/-}$ mice received a single injection of AAV-GAA at $1\times10^{11}$ vg/kg with or without a regimen of 1-Deoxynojirimycin (DNJ) and ambroxol (ABX) given at 3 days on followed by 3 days off intervals. Untreated $Gaa^{+/+}Cd4^{-/-}$ and $Gaa^{-/-}Cd4^{-/-}$ mice were used as controls. **B** Analysis of GAA activity in blood. **C** Western blot analysis of circulating GAA protein. **D** Quantification of circulating GAA protein bands in **B**. Analysis of GAA activity (**E**) and glycogen content (**F**) in muscles at the end of the study. **G** Heart weight expressed as percentage of body weight. **H** Analysis of grip strength 2 months after treatment. Data shown as average ± SD. Statistical analysis: **B** Two-way ANOVA with Tukey post-hoc test; **D** $t$-test; **E–H** One-way ANOVA with Tukey post-hoc test *, † and # $p < 0.05$; **, †† and ## $p < 0.01$; ***, ††† and ### $p < 0.001$; ****, †††† and #### $p < 0.0001$. Exact $p$ values for **B**, **E–H** are provided in the Source Data file. **B–H**: $n = 8$ mice per treatment group.

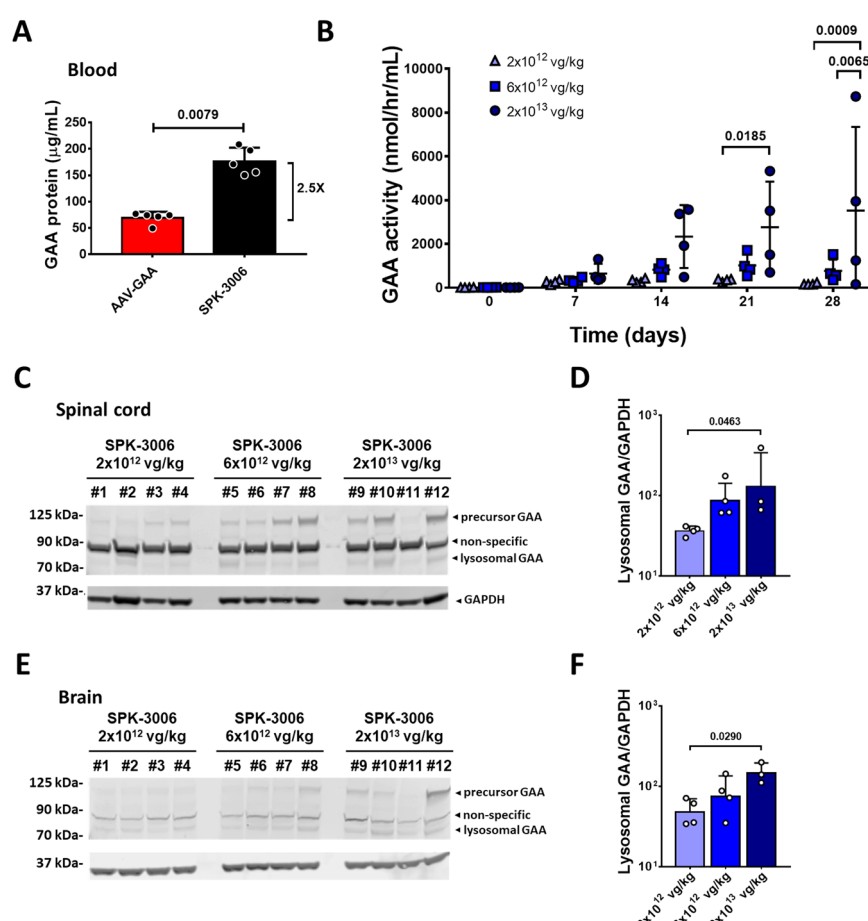

**Fig. 9 SPK-3006 drives vector dose-dependent expression of GAA in liver and tissue uptake in non-human primates. A** Circulating GAA antigen levels 4 weeks of post-injection of C57BL/6 mice with AAV-GAA or SPK-3006 at $2\times10^{12}$ vg/kg ($n = 5$ per group). **B** GAA activity in plasma of non-human primates treated with SPK-3006 ($n = 4$ per group). **C, E** Western blot analysis of GAA antigen levels in spinal cord (**C**) and brain (**E**). **D, F** Quantification of lysosomal GAA antigen in spinal cord (**D**) and brain (**F**) ($2\times10^{12}$ and $6\times10^{12}$ vg/kg, $n = 4$ per group; $2\times10^{13}$, $n = 3$). Data shown as average ± SD. Statistical analysis: **A** Two-sided Mann–Whitney $t$-test; **B** Two-way ANOVA with Tukey post-hoc test; **D, F** One-way ANOVA.

Secretion of GAA from hepatocytes into the circulation was confirmed by the analysis of GAA activity in plasma (Fig. 9B). Levels of GAA activity in plasma were dose-dependent and consistent with the VGCN measured in the liver at sacrifice (Fig. 9B and Supplementary Fig. 6B). Lysosomal GAA in tissues harvested at day 28 was then analyzed by Western blot. One animal in the high-dose cohort (#11, Supplementary Table 1) was excluded from the statistical analyses due to unexplained lower levels of transgene expression, not justified by the development of anti-hGAA IgG (Supplementary Table 1) or by a lower VGCN in liver (Supplementary Fig. 6B). In agreement with the GAA activity measured in plasma (Fig. 9B), we detected a dose-dependent increase of lysosomal GAA in heart, diaphragm, and skeletal muscle (Supplementary Fig. 7). We also detected a dose-dependent accumulation of lysosomal GAA in spinal cord and brain (Fig. 9C–F). Analysis of vector biodistribution in multiple

tissues showed that most vector genome copies were found in liver and spleen (Supplementary Fig. 8). Analysis of liver transaminases and additional clinical biochemistry parameters supported the safety of the SPK-3006 vector at all the doses tested (Supplementary Table 2).

These results support the safety and feasibility of expressing secretable GAA in liver to target peripheral tissues. Importantly, our data suggests that the stable amounts of circulating GAA achieved through efficient liver gene transfer may facilitate enzyme uptake through the blood-brain barrier.

## Discussion
Despite being the only therapeutic approach currently available for the management of PD, ERT suffers from several limitations. PD remains lethal for a relatively large proportion of IOPD and

severe LOPD patients[40–42]. Additionally, in ERT-responsive patients, disease progression and CNS manifestations have been reported[4,9–14,43]. We previously showed that liver expression of a secretable form of GAA via AAV gene therapy in mice can potentially treat[21] or reverse[22] PD, even in severe models of the pathology[44].

Here, we demonstrated that gene transfer with secretable GAA holds enhanced therapeutic potential when compared to standard of care ERT. We provided mechanistic insights into the pharmacokinetics of liver-expressed GAA, showing that steady-state levels of enzyme in the circulation attainable with hepatic gene therapy provide greater bioavailability in refractory tissues compared to ERT. We also showed that combination of gene therapy with pharmacological chaperones results in enhanced therapeutic efficacy, further confirming that increasing the circulating levels of hepatocyte-secreted GAA provides better therapeutic efficacy. Last, we provided evidence of safety and tissue uptake of secretable GAA in a large animal model of gene transfer.

We generated a colony of immunodeficient $Gaa^{-/-} Cd4^{-/-}$ mice and demonstrated therapeutic superiority of liver gene transfer with secretable GAA vs. ERT with rhGAA. While ERT was as effective as gene transfer in clearing glycogen from heart, a tissue permissive to rhGAA uptake[35], in refractory tissues like skeletal muscle[35] ERT did not clear glycogen significantly. These results are consistent with emerging data from the clinic, which suggest that more frequent infusions of rhGAA at doses higher than the standard of care are more efficacious in treating PD[45], underpinning the hypothesis that uptake of the recombinant enzyme in refractory tissues is limited. The current study did not test more intense dose regimens for ERT, based on the assumption that they are not easily sustainable clinically, in particular in LOPD patients, due to the high cost of rhGAA and the treatment burden for patients associated with frequent and lengthy infusions[46].

In the setting of hepatic gene transfer with secretable GAA in mice, we demonstrated correction of glycogen accumulation, several ultrastructural and biochemical markers of lysosomal and mitochondrial function, and strength in muscle at vector doses as low as $1 \times 10^{11}$ vg/kg. Additionally, we also show partial correction of glycogen accumulation in the CNS of treated mice. These results were further supported by the preliminary evidence of tissue accumulation of the lysosomal form of GAA[7] in non-human primates in a 28-day study. These data are encouraging, as they demonstrate that liver expression of GAA, even at low levels, can be therapeutically beneficial. Further, based on our previous observations on the dose-dependence and time-dependence of GAA accumulation and glycogen clearance in tissues in the setting of gene transfer[21,22], it is possible that our study underestimates the full therapeutic potential of liver expression of GAA over the long term. Similarly, the effect of secretable GAA on lysosomal function normalization, and potentially on other lysosomal enzymes[47], remains to be assessed. The safety of liver expression of secretable GAA in non-human primates is also worth noting, as the lack of significant alterations of clinical parameters, along with the ability of achieving body-wide uptake of GAA, is encouraging from a translational point of view. Importantly, when comparing the liver approach shown here with direct muscle targeting with gene transfer, one potential additional layer of safety comes from the lower predicted therapeutic AAV vector doses. Direct targeting of muscle with systemic delivery of AAV vectors does, in fact, require vector doses in excess of $1 \times 10^{14}$ vg/kg[48,49] (NCT03199469, NCT03368742, and NCT03375164), which in some cases have triggered complement activation and severe acute toxicities in humans[50]. High vector doses are also associated with a higher risk of development of detrimental cytotoxic immune responses against AAV transduced cells[51]. The use of hepatotropic capsids[39,52] can potentially help decrease the overall risk of immune mediated toxicities.

An intriguing aspect of our study is that the modeling of the pharmacokinetics of GAA expressed as a transgene in liver, compared to that of ERT with rhGAA, fully supports the efficacy data in mice. ERT is characterized by peaks of enzyme activity followed by troughs, which result in complete disappearance of the enzyme from tissues few days after treatment. This leads to periods between rhGAA infusions in which glycogen can continue to accumulate in tissues and is associated with low rescue of refractory tissues like skeletal muscle, in which enzyme uptake is limited. In the setting of gene transfer, steady-state supply of secretable GAA through the bloodstream provides a continuous inflow of enzyme into the tissue and consequent better clearance of glycogen. The calculation of the AUC of GAA for ERT and AAV gene transfer fully supports this model, showing an identical bioavailability of GAA in heart and significant differences in favor of AAV gene transfer in skeletal muscle. The differential uptake of GAA across tissues has been previously described[35,53] and correlates with the clinical findings, as heart manifestations of PD are usually rapidly reversed by ERT[40], while skeletal muscle manifestation are generally harder to correct[15,16]. The full explanation of the mechanism(s) underlying these observations is lacking, although one possibility is that the expression levels of M6PR across tissues could drive the differential uptake of the enzyme[54]. To this end, the continuous availability of GAA driven by the hepatocyte expression (vs. the limited temporal persistence of rhGAA in the bloodstream following ERT) may help overcome limitations related to the ability of a tissue to take up the enzyme.

Our findings on the pharmacokinetics of AAV expression of GAA are broadly relevant to the field of gene therapy, as they provide mechanistic insights and a framework for the evaluation of the efficacy of gene therapy vs. existing protein replacement therapies. Specifically, they suggest that the target therapeutic levels extrapolated from conventional protein therapies may not directly translate to gene therapy. An example of the favorable pharmacokinetics offered by gene therapy comes from liver trials for hemophilia[39,55,56], showing an almost complete correction of the bleeding diathesis in patients over a broad range of clotting factor levels, even lower than those predicted based on protein replacement therapy[57]. While the context of hemophilia differs from that of Pompe disease, as clotting factors act directly in blood and not in peripheral tissues, human trial data show that steady state levels of transgene expression positively affect clinical outcomes. Thus, the superior pharmacokinetics of liver-expressed protein therapeutics, demonstrated here, can potentially offer opportunities to develop more efficacious therapeutic interventions across a broad range of indications[58].

ERT approaches aimed at increasing the bioavailability of rhGAA are based on the use of PCs, i.e., molecules that increase the stability of the enzyme[17–19,37,38,59]. Here, we provide evidence that a similar approach could be applied to gene therapy, as a combination of PCs[36] with expression of secretable GAA in liver resulted in improved glycogen clearance in the relevant tissues. The enhanced therapeutic efficacy obtained with PCs in mice further support the hypothesis that sustained, stable levels of circulating GAA efficiently rescue PD. While the need to supplement gene therapy with PCs will have to be evaluated in the clinic, the combination of gene therapy with pharmacological approaches to enhance efficacy is an important concept that has previously been explored, for example, in mice in the context of autoimmunity[60] and in clinical trials of gene therapy for spinal muscular atrophy[61]. In PD gene therapy, the use of PCs can provide a potential avenue to reduce the therapeutic dose of

vector or to boost enzyme activity in tissues in case of sub-therapeutic levels of expression of the GAA transgene.

Clearance of glycogen from the diaphragm has been long considered important to rescue respiratory symptoms of PD. To this aim, a diaphragm-directed gene therapy trial has been previously conducted[62]. Notably, diaphragm stimulation defects have been reported in PD subjects[63], suggesting that the correction of the pathological consequences of glycogen accumulation in this muscle would be not sufficient to improve the respiratory function in the absence of concomitant correction of afferent phrenic motor neurons in the spinal cord. Here, we transiently detected lysosomal GAA in the spinal cord of animals treated with ERT, although only animals receiving gene transfer at the higher AAV vector doses showed a significant reduction in glycogen in CNS. Leakage across the BBB[64], transport via exosomes[65], or lysosomal exocytosis at neuromuscular junctions[66], could contribute to the transport of GAA from the systemic circulation to the CNS. Although preliminary, the results obtained in non-human primates suggest that stable liver expression of secretable GAA drives CNS uptake of the enzyme. Future studies in which animals are followed for a longer time after gene transfer may help strengthen the data shown here. Moreover, additional data coming from long-term follow-up of PD patients will strengthen the link between glycogen accumulation in lysosomes and CNS pathology.

One of the limitations of hepatic gene transfer with AAV is that, given the episomal nature of AAV vector genomes, persistence of transgene expression in neonatal animal models cannot easily be achieved due to hepatocyte proliferation[67,68]. AAV vector administration results in long-term development of high titer neutralizing antibodies to the vector capsid[69], making redosing difficult. In view of developing a gene therapy approach based on secretable GAA for IOPD, one possible solution is to use strategies that can allow to overcome the limitation of anti-AAV antibodies[70–72]. Alternatively, we showed that the co-expression of secretable GAA in liver and tissues like muscle, less prone to proliferation during growth, allow for long-term transgene expression also in neonate animals[73].

Last, differently from ERT, hepatic AAV gene transfer has been shown to induce immune tolerance to the encoded transgene products[74], including GAA[23,75]. We have previously reported that the engineered secretable GAA proteins are less immunogenic than the native protein form[21], possibly via more efficient induction of central tolerance, as observed with other model antigens[76], and that dominant liver expression prevents transgene immune responses[73,77]. Thus, one potential additional advantage of liver-expressed secretable GAA, broadly compared to ERT, is its beneficial therapeutic transgene immunogenicity profile.

In conclusion, we provided a model to study the pharmacokinetics of GAA in mice subjected to ERT or AAV gene transfer to the liver. Our data indicate that gene transfer provides superior enzyme bioavailability to refractory tissues, including CNS, in mice, resulting in superior therapeutic efficacy compared to ERT. Pharmacological chaperones can enhance the efficacy of gene transfer, further supporting the central role of steady-state enzyme uptake in tissues in driving efficient glycogen clearance. Preclinical optimization of the gene therapy approach and scale up of the findings to non-human primates support the clinical translation of the approach in PD.

## Methods

**AAV vector production.** AAV8 vectors encoding for a secretable form of human GAA, also known as AAV8-hAAT-sp7-Δ8-coGAA[21] were produced by standard triple transfection[78]. The AAV vector used in non-human primates (SPK-3006) was composed of the liver-tropic Spark100 capsid[39] and a codon-optimized secretable GAA cDNA.

**In vivo studies.** In vivo studies were performed in compliance with all relevant ethical regulations for animal testing and research.

**Mouse models.** Comparative efficacy and pharmacokinetics studies between gene therapy and ERT were performed in male $Gaa^{-/-}Cd4^{-/-}$ mice of two months of age. Male and female $Gaa^{-/-}$ mice (B6;129-$Gaa^{tm1Rabn}$/J stock no. 004154) and $Cd4^{-/-}$ mice (B6.129S2-$Cd4^{tm1Mak}$/J stock no. 002663) were purchased from the Jackson Laboratory and crossed for the generation of $Gaa^{-/-}Cd4^{-/-}$ mice (Supplementary Fig. 1A). The Gaa and Cd4 genotype was confirmed by PCR on genomic DNA with oligonucleotides specific for the mutated regions in each gene (Supplementary Table 3). C57BL/6 mice (8–10 weeks of age, Charles River Laboratories) were used to compare the optimized SPK-3006 with AAV8-hAAT-sp7-Δ8-coGAA.

**In vivo studies in mice.** In all mouse studies, animals were randomly assigned to treatment groups. To minimize potential bias during functional assessments in mice, operators were blinded to the study design. Operators in charge of sample analysis were blinded to study design.

Treatment with pharmacological chaperones (PCs) was performed in male $Gaa^{-/-}Cd4^{-/-}$ mice of 4 months of age that received either vehicle or duvoglustat-HCl (DNJ, Interchim, San Diego, CA) at 100 mg/kg per day combined with ambroxol-HCl (ABX, Sigma-Aldrich, Saint-Louis, MO) at 25 mg/kg per day dissolved in drinking water following a "3 days on /4 days off" regimen as previously described[59] for 66 days. One day after the start of the chaperone treatment, vehicle (PBS) or AAV8-hAAT-sp7-Δ8-coGAA were administered at $1 \times 10^{11}$ vg/kg via tail vein.

rhGAA was administered via retro-orbital injection to anesthetized mice together with an intraperitoneal injection of the anti-histaminic drug diphenhydramine hydrochloride (Sigma-Aldrich) at 25 mg/kg to further prevent the risk of anaphylaxis.

Age-matched and sex-matched $Gaa^{-/-}Cd4^{-/-}$ and $Gaa^{+/+}Cd4^{-/-}$ littermates were used as healthy controls in the studies.

Male C57BL/6 mice (8–10 weeks of age, Charles River Laboratories) were used to compare the optimized SPK-3006 with AAV8-hAAT-sp7-Δ8-coGAA. Five C57BL/6 animals per group were injected via tail vein infusion with $2 \times 10^{12}$ vg/kg of either vector and plasma samples were drawn on week 3 to assess hGAA antigen levels. The AAV8-hAAT-sp7-Δ8-coGAA vector was administered intravenously to awake, restrained animals via tail vein injection.

Mouse studies were performed according to the French and European legislation on animal care and experimentation (2010/63/EU) and approved by Genethon's ethical committee (protocol n° 2017-011-B #13643).

**In vivo studies in non-human primates.** For NHP studies, animals were pre-screened for anti-AAV capsid neutralizing antibodies previous to vector administration. SPK-3006 was administered to anesthetized seronegative male Rhesus macaques (Macaca mulatta, $n = 4$ per dose, 3–5 years of age) via saphenous vein injection at the doses of $2 \times 10^{12}$, $6 \times 10^{12}$, and $2 \times 10^{13}$ vg/kg, animals were randomly assigned to treatment groups. Operators in charge of sample analysis were blinded to study design. NHPs were monitored for GAA activity and antigen levels in blood and tissues, clinical observations, body weight, and clinical chemistry over the course of 4 weeks. In vivo procedures were conducted at Charles River (Mattawan, MI, United States), according to the Animal Welfare Act (Title 7 United States Code, Sections 2131-2159) and the Animal Welfare Regulations (Title 9 Code of Federal Regulation, Parts 1, 2, and 3). The study was approved by the Institutional Animal Care and Use Committee (IACUC) of Charles River Mattawan (protocol n° 2377-015, approved on 23/04/2018).

**Blood GAA activity assay and anti-GAA IgG measurement.** Snap-frozen tissues were homogenized in UltraPure DNase/RNase-free water (Thermo Fisher Scientific, Waltham, MA) with FastPrep lysis tubes (MP Biomedicals, Ohio, USA), followed by centrifugation 20 min at 10,000×g to collect the supernatant. Protein content in lysates was quantified by BCA Protein Assay (Thermo Fisher Scientific). Blood samples were collected by retro-orbital sampling with heparinized capillary tubes and mixed with 3.8% w/v sodium citrate, followed by plasma isolation. GAA activity measurement was performed as already described[21]. The concentration of anti-hGAA IgG antibodies in mouse and NHP plasma was determined by enzyme-linked immunosorbent assay (ELISA)[21].

**Western blot analyses.** Tissue homogenates and plasma samples were prepared as described for the analysis of GAA activity. SDS-page electrophoresis was performed with NuPAGE 4–12% Bis-Tris protein gels (Life technologies, Carlsbad, CA). After transfer, membranes were blocked with Odyssey buffer (Li-Cor Biosciences, Lincoln, NE) and incubated with primary antibodies: anti-human GAA (rabbit monoclonal, dilution 1:1000, Abcam, Cambridge, MA; clone EPR4716), anti-mouse SQSTM1/p62 (mouse monoclonal, dilution 1:1000, Abcam), anti-mouse PARK2/parkin (rabbit polyclonal, dilution 1:1000, Proteintech, Rosemont, IL), anti-mouse GAPDH (mouse polyclonal, PA1-988, dilution 1:500, Thermo Fisher Scientific), anti-mouse α-tubulin (mouse monoclonal, clone DM1A, dilution 1:500, Sigma-Aldrich) or anti-mouse vinculin (mouse monoclonal, clone hVIN-1, dilution 1:250, Sigma-Aldrich). Membranes were then washed and incubated with the appropriate secondary antibody (Li-Cor

Biosciences) and visualized by Odyssey imaging system (Li-Cor Biosciences). The concentration of hGAA transgene product in plasma of C57BL/6 mice was determined using the Protein Simple WES/Sally Sue platform (Bio-Techne, Minneapolis, MN), a capillary electrophoresis immunoassay, according to the manufacturer's instructions. hGAA concentration was determined against a standard curve made with serial dilution of rhGAA. Densitometry analysis was conducted using Compass® software (Bio-Techne) version 5.0.1.

**Vector genome copy number**. Vector genome copies in mice and NHPs were determined by qPCR on total tissue DNA. Total DNA was extracted from liver homogenates with the Qiagen DNA extraction method according to manufacturer's instructions. The number of vector copies per diploid genome was determined using specific primers to amplify the GAA transgene sequence (Supplementary Table 3). The number of vector copies was normalized by the copies of the *titin* gene, which was used as internal control for each sample (Supplementary Table 3). Data were expressed as vector genome copies per diploid genome.

**Glycogen assay and muscle ROS analysis**. Glycogen assay was performed as already described[21]. ROS production was evaluated by EPR[22]. Snap-frozen *tibialis anterior* samples were used for the analysis. Results were expressed in arbitrary units and normalized for microgram of protein in the assay.

**Grip strength**. Muscle strength was assessed using a grip strength meter (Columbus Instruments, Columbus, OH)[79]. Briefly, mice were lifted by the tail to the same height of the grip strength meter grid. Mice were then moved horizontally until they were within reach. Four-limb grip was inspected visually to confirm the symmetry and the tight grip. Mice were then gently pulled away from the grid until the grasp was released. Mean values of three independent measures expressed in Newtons were reported.

**Lysosomal length determination and fiber size measurement**. For lysosomal length determination, fresh *tibialis anterior* muscle was collected and processed for electron microscopy[22]. For the quantification of fiber size, quadriceps were snap-frozen in isopentane previously chilled in liquid nitrogen. Serial 8 µm cross sections were cut with a Leica CM3050 S cryostat (Leica Biosystems). Quadriceps sections were blocked with 10% normal goat serum (NGS) in PBS for 30 min at room temperature. Then, sections were stained with an anti-mouse laminin primary antibody (rabbit polyclonal, dilution 1:400, Dako Agilent, Santa Clara, CA), followed by incubation with a secondary antibody goat anti-rabbit Alexa Fluor 594 (Thermo Fisher Scientific). Cell nuclei were stained DAPI fluoromount-G mounting medium (Southern Biotech, Birmingham, AL). Representative images were acquired with a slide scanner Axio Scan.Z1 with ×10 magnification (Carl Zeiss, Oberkochen, Germany). For the determination of the minimal and maximal diameter of fibers, laminin staining was used as described above to delimit each fiber. Images were then analyzed with the help of ImageJ software version 1.51. Diameters were taken from all fibers present in one section from three animals per group (one section per animal). The data shown represents the average from each section ($n = 3$).

**Statistics**. All data are expressed as mean ± SD. GraphPad Prism 7 software (GraphPad Software) was used for the statistical analyses. $p$-value < 0.05 was considered significant. The number of sample unit ($n$), upon which we reported statistics is the single animal. Parametric tests were used for data having a normal distribution with $\alpha = 0.05$. One- and Two-way ANOVA with Tukey's post-hoc correction were used for comparisons between more than two groups and one or two variables, respectively. Two-tailed $t$-test was used for two-group comparisons with normally distributed data: Student's $t$-test (equal variance data), Welsh's $t$-test (unequal variance data). Non-parametric Mann–Whitney test was used for two-group comparisons with non-normally distributed data. The statistical analysis performed for each data set is indicated in the corresponding figure legend.

**Reporting summary**. Further information on research design is available in the Nature Research Reporting Summary linked to this article.

## Data availability

All data presented in the manuscript are available in the Source Data file. Source data are provided with this paper.

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

## Acknowledgements

This work was supported by Genethon and the French Muscular Dystrophy Association, the European Union's research and innovation program under grant agreement no. 667751 (to F.M.), the European Research Council Consolidator Grant under grant agreement no. 617432 (to F.M.), Marie Skłodowska-Curie Actions Individual Fellowship (MSCA-IF) grant agreement no. 797144 (to U.C.), a grant from the DIM Thérapie Génique (to D.A.G.) and by Spark Therapeutics under a sponsored research agreement to Genethon. We thank Nicolas Guerchet and Guillaume Tanniou for help with the functional testing; Emilie Bertil-Froideveaux, Fanny Bordier and the Genethon Histology Core for technical help with histology; Simon Jimenez and the Genethon Imaging Core for technical help with histology image acquisition. We also thank the Imaging and Cytometry Core Facility of Genethon and the Bioanalytical and Pre-Clinical Manufacturing Operation teams at Spark Therapeutics.

## Author contributions

H.C.V., F.C., and C.R.R. performed or directed experimental activities, contributed significantly to experimental design, data analysis, and manuscript writing. S.M.R., P.C., G.R., and F.M. conceived the study, directed experimental activities and data interpretation and wrote the manuscript. D.G., O.B., and X.M.A. contributed to experimental design and results interpretation. P.S., J.M.L.N., G.M.P., J.F., S.B., U.C., M.M.N., J.K.L., G.T., N.G., L.v.W., N.D., B.G., J.C., S.J., C.A., M.S.S., S.C., M.L., M.C., T.A., and W.J.Q.III contributed to experimental activities. All authors approved the final manuscript.

## Competing interests

F.M., P.C., and G.R. are inventors in patents applications concerning the treatment of Pompe disease by AAV licensed to Spark Therapeutics (WO2018046774, WO2018046775, and WO2018046775). S.M.A., J.M.L.N., and X.M.A. are inventors in a patent application describing the optimization of the secretable GAA transgene owned by Spark Therapeutics (WO2019222411). C.R.R., J.M.L.N, G.M.P., M.L., M.C., T.A., W.J.Q.III, X.M.A., S.M.A., and F.M. are either current or past employees and equity holder of Spark Therapeutics, Inc., a Roche company. H.C.V., P.S., U.C., J.F., S.B., M.N.-N., J.K.-L., L.v.W., N.D., B.G., J.C., C.A., M.S.-S., S.C., D.G., and O.B. declare no competing interests. This work was partially supported by Spark Therapeutics under a sponsored research agreement to Genethon.
