## [Peer Review File · Nature Communications]

Reviewers' Comments:

Reviewer #1:

Remarks to the Author:

This manuscript describes the development of a gene therapy approach to the treatment of Pompe disease in a mouse model of CRIM negative infantile Pompe disease and in non-human primates. It describes a concept of gene therapy which is currently prevalent in the lysosomal storage disorders where the aim is not to deliver a functional gene directly to the target cells but instead to transduce other cell types with the aim of getting them to overexpress a secretable form of the defective enzyme which then enters the circulation and is taken up by the affected cell types. In this approach, the transducer cells, which can be hepatocytes or bone marrow progenitors, act as an enzyme factory to maintain constant enzyme levels in the circulation. The hope is that this will prove more effective than current enzyme replacement therapies where recombinant enzyme is given by periodic intravenous infusion.

This is not a novel approach and is under investigation by many groups in many different LSDs. In fact, clinical trials of exactly this approach in Pompe disease are already recruiting (<https://clinicaltrials.gov/ct2/show/NCT03533673?recrs=a&cond=Pompe+Disease&draw=2&rank=5>, <https://clinicaltrials.gov/ct2/show/NCT04174105?recrs=a&cond=Pompe+Disease&draw=2&rank=10>) including one using what I think is the vector described here (<https://clinicaltrials.gov/ct2/show/NCT04093349?recrs=a&cond=Pompe+Disease&draw=3&rank=19>). I think the authors need to take more care to set out this context in the introduction.

Although therapeutic efficacy of AAV-mediated GAA gene delivery to the liver has been demonstrated before by this group and others, what is novel here is the comparison with ERT. In order to do this, they have had to construct a novel immunodeficient mouse model which avoids the development of hypersensitivity to the infused enzyme. It needs to be made clear that this model does not really equate to any potential human use of the technology, and that the distribution and pharmacokinetics of both the recombinant and hepatic expressed enzymes are likely to be affected by the immunodeficiency. Nonetheless, the results are interesting and likely to be relevant to human use.

The primary results concern the comparison between the pharmacokinetics of ERT and gene therapy. I think figure 1B, showing plasma enzyme activity, is misrepresentative as it has no data from early time points after enzyme infusion, when activity after ERT would have been high. This data is present in figure 6 and I would suggest that at least the data in fig 6B is included in fig 1.

Fig 2A is confusing because the scales on the y axis are very different for each panel. I would suggest it needs an extra panel to indicate the enormous differences in the level of enzyme seen in different muscles. I would suggest comparing highest dose expression across the different tissues. These disparities then need to be picked up in the text and discussed.

I have some problems with the sections on enzyme delivery to the CNS. Firstly CNS disease is not a significant feature in the natural history of Pompe disease; it has recently come to light as a late complication in patients with IOPD who have been successfully treated with ERT, which is in fact lifesaving in this condition. To suggest that delivery to the CNS is a major unmet clinical need is inaccurate. Secondly, there is no satisfactory explanation of how enzyme is accessing the CNS. It is widely accepted that lysosomal enzymes in the blood do not cross the blood brain barrier. The data presented here shown that enzyme can enter endothelial cells, which has already been well demonstrated in other LSDs, but not that it can be exported again into the brain. This data is a misleading distraction and should be omitted. Instead they should consider other explanations: either that enough vector genomes enter the CNS to lead to significant gene expression (there does not appear to be any data here about where vector genomes are found in mice, which would be useful, at least in the supplementary material); or that expression is from bone marrow derived cells (where they have found significant vector genomes in NHP) repopulate the CNS, which is a well recognised mechanism.

I am not sure what the added value of the chaperone work is. If the liver directed gene therapy

approach works so well, then why would we need a further, expensive, pharmacological intervention. They have no evidence that this would provide added clinical benefits. Without that I don't think it adds anything to the manuscript.

In the Discussion, the comparison with haemophilia (line 432) is not really appropriate as factor VIII is a protein which is normally secreted from the liver and acts in plasma rather than having to be taken up to cells: delivering the factor VIII gene to the liver in haemophilia is actually correcting the affected tissue, which is not what is being proposed here at all. The sentence about CNS glycogen storage having an important role in respiratory failure in Pomoe (starting on line 450) is not evidence based and should be omitted. This is part of the authors unjustified attempt to portray CNS delivery of enzyme as an unmet need. Their data on muscle delivery is good enough, and they really don't need to make doubtful claims about CNS delivery.

Reviewer #2:

Remarks to the Author:

The manuscript NCOMMS-21-06438, entitled "Hepatic expression of GAA results in enhanced enzyme bioavailability in mice and non-human primates" by Mingozi and colleagues, reports on hepatic gene transfer with adeno-associated virus vectors expressing secretable GAA.

It is a very thorough study that is well described and elegantly conducted with state-of-the-art methods.

The investigation with a Pompe disease mouse model reveals great therapeutic potential for hepatic gene transfer with AAV expressing secretable GAA. This finding should raise interest in the field of lysosomal storage diseases. The strength of the work is the careful comparison of the corrections brought about by the AAV intervention with those reached with enzyme replacement therapy (the present therapy for Pompe disease). Of interest is also the reported examination of the potential additive value of administration of small compound chaperones to AAV treated Pompe mice.

Specific comment to authors.

The study is well conducted with attention to detail.

One wonders whether it is not worthwhile to also consider an analysis of unrelated lysosomal enzyme levels in tissues of treated and untreated PD mice. Often in LSDs, other lysosomal enzymes are secondary elevated in response to lysosomal storage. A successful clearance of excessive glycogen by the AAV intervention would likely be accompanied by corrections in other lysosomal enzymes as well. Demonstration of such corrections would further demonstrate efficacy of the AAV intervention.

REVIEWER COMMENTS

Reviewer #1 (Remarks to the Author):

This manuscript describes the development of a gene therapy approach to the treatment of Pompe disease in a mouse model of CRIM negative infantile Pompe disease and in non-human primates. It describes a concept of gene therapy which is currently prevalent in the lysosomal storage disorders where the aim is not to deliver a functional gene directly to the target cells but instead to transduce other cell types with the aim of getting them to overexpress a secretable form of the defective enzyme which then enters the circulation and is taken up by the affected cell types. In this approach, the transducer cells, which can be hepatocytes or bone marrow progenitors, act as an enzyme factory to maintain constant enzyme levels in the circulation. The hope is that this will prove more effective than current enzyme replacement therapies where recombinant enzyme is given by periodic intravenous infusion.

ANSWER: We would like to thank the Reviewer for appreciating the relevance of our work and for helping us improve our manuscript.

This is not a novel approach and is under investigation by many groups in many different LSDs. In fact, clinical trials of exactly this approach in Pompe disease are already recruiting (<https://clinicaltrials.gov/ct2/show/NCT03533673?recrs=a&cond=Pompe+Disease&draw=2&rank=5>, <https://clinicaltrials.gov/ct2/show/NCT04174105?recrs=a&cond=Pompe+Disease&draw=2&rank=10>) including one using what I think is the vector described here (<https://clinicaltrials.gov/ct2/show/NCT04093349?recrs=a&cond=Pompe+Disease&draw=3&rank=19>). I think the authors need to take more care to set out this context in the introduction.

ANSWER: The Reviewer correctly states that many groups are exploring the use of the liver as a biofactory to secrete proteins. It is also correct that the approach described in our manuscript is currently under clinical investigation. There are several points supporting the novelty of our work that we would like to highlight:

1. our work provides a model to study the pharmacokinetics of gene transfer and compare that to ERT. While we do not disagree that other studies showed that gene therapy is superior to ERT, here we provide mechanistic insights explaining the findings, which are relevant to several of the gene therapies in development. We also show that ERT may have the same efficacy as gene transfer in permissive tissue (e.g. heart), while enzyme bioavailability in refractory tissues is higher in AAV treated animals. This has direct implications for Pompe and other diseases and provides a framework for the evaluation of efficacy and minimal efficacious dose of gene therapy for liver-expressed therapeutic proteins.

2. This study is the first study in which the pharmacokinetics of an enzyme secreted in hepatocytes has been evaluated in the circulation and in tissues. The work provides direct evidence that the steady-state levels of expression drive

therapeutic efficacy, and that peak levels of enzyme achieved with ERT are essentially irrelevant when it comes to long-term outcomes. Highlighting these differences has important implications for gene therapy trial design, definition of the starting dose of vector, and evaluation of therapeutic efficacy of gene therapy vs. ERT. Importantly, we believe these are findings that are relevant to virtually all trials in which a secreted protein is expressed, they also highlight that the pharmacology data developed in the context of protein therapy may not be fully relevant to gene therapy.

3. the data we presented on the combination of chaperones with gene transfer is also new and relevant to Pompe disease and other LSDs (see comment below). We strongly believe that presenting these data to the scientific community will support the concept (explored in the clinical context, see below) that combination of gene therapy with other pharmacological therapies may result in better therapeutic outcomes.

We capture these considerations in the amended version of our manuscript.

Introduction, we acknowledge the fact that there are clinical trials exploring the approach presented:

“Gene therapy with adeno-associated virus (AAV) vectors provides the opportunity to turn the liver into a biofactory for the expression of therapeutic proteins, a concept demonstrated in preclinical studies ^{1, 2} and currently explored in clinical trials (e.g. ClinicalTrials.gov NCT03533673, NCT04093349).”

We also added the ClinicalTrial.gov ID at the end of the introduction to better highlight that the approach discussed is currently being explored in the clinic.

Discussion, several paragraphs point to the novelty of the work. Nevertheless, we added a paragraph to better highlight the above considerations:

“Our findings on the pharmacokinetics of AAV expression of GAA are broadly applicable to the field of gene therapy, as they provide a framework for the evaluation of the efficacy of gene therapy vs. existing protein replacement therapies. Specifically, they suggest that the target therapeutic levels extrapolated from conventional protein therapies may not directly translate to gene therapy.”

Although therapeutic efficacy of AAV-mediated GAA gene delivery to the liver has been demonstrated before by this group and others, what is novel here is the comparison with ERT. In order to do this, they have had to construct a novel immunodeficient mouse model which avoids the development of hypersensitivity to the infused enzyme. It needs to be made clear that this model does not really equate to any potential human use of the technology, and that the distribution and pharmacokinetics of both the recombinant and hepatic expressed enzymes are likely to be affected by the immunodeficiency. Nonetheless, the results are interesting and likely to be relevant to human use.

ANSWER: We thank the Reviewer for the feedback on the interest and relevance of our work. We also understand the potential concern related to the use of an immunodeficient mouse model of Pompe disease. In our manuscript, we characterized the novel mouse colony and confirmed both glycogen accumulation in several tissues and decrease in muscle strength, key feature of the disease. Like

others, we previously showed that Pompe mice exhibit neuroinflammation^{2, 3}, and breeding of Pompe mice in a background that exacerbates inflammation leads to worsened phenotype⁴. While the immunodeficient mouse presented here is unlikely to have the same immunological features of immunocompetent Pompe mice, we do not have any evidence of an impact on the pharmacokinetics and biodistribution of the liver-expressed GAA. Previous studies from us² and others^{5, 6} demonstrated that hepatic expression of GAA results in suppression of anti-GAA antibody production, therefore the immunodeficient Pompe mouse is unlikely to display different pharmacokinetics and biodistribution of GAA in the context of gene therapy. Of note, the immune system is not entirely "shut down" in CD4 KO mice as they show normal development of CD8+ T cells and myeloid compartment is unaltered⁷. In the context of ERT, repeated administration of rhGAA results in rapid development of anti-GAA antibodies in Pompe mice, thus requiring concomitant immunosuppression⁸. Thus, the immunodeficient mouse, while not perfect, is a better representation of the human context, at least when it comes to adult Pompe patients in which anti-GAA antibodies are observed in a minority of subjects and do not seem to have a major impact on GAA efficacy⁹.

We capture these considerations in the revised version of our manuscript, Results section, by clarifying the evaluations that were conducted in the immunodeficient Pompe colony and potential limitations of the model related to immunodeficiency:

"Analyses of wild-type ($Gaa^{+/+}Cd4^{-/-}$) and affected ($Gaa^{-/-}Cd4^{-/-}$) mice at 4 months of age confirmed the **accumulation of glycogen in several tissues and the decrease in muscle strength** in the colony (Supplementary Fig. 1B,C). As previously observed in PD mice, no significant alterations of respiratory function were detected in 4-month-old mice (Supplementary Fig. 1D). **Due to the immunodeficient background, $Gaa^{-/-}Cd4^{-/-}$ mice were not expected to present the proinflammatory features previously described in the original colony^{2, 3}.**"

The primary results concern the comparison between the pharmacokinetics of ERT and gene therapy. I think figure 1B, showing plasma enzyme activity, is misrepresentative as it has no data from early time points after enzyme infusion, when activity after ERT would have been high. This data is present in figure 6 and I would suggest that at least the data in fig 6B is included in fig 1.

ANSWER: We agree with the Reviewer that having peak levels of rhGAA in the context of ERT could be informative. We have amended Figure 1 according to what suggested. **Panel 6B is now Panel 1C and it is described in the Results section.**

We would like to clarify one important point about the data. While we agree that it is useful to see the blood kinetics of GAA activity after ERT, the data in plasma collected at through shown in Figure 1B are also important to understand the difference in mechanism of action of ERT vs. gene therapy: one is based on the infusion of a bolus of recombinant enzyme which persists transiently, the other is based on the steady-state expression of GAA with no peak and through.

We highlighted this difference in the Results section:

"Overall, these results show that, in Pompe mice over an 18-week follow-up, hepatic expression of secretable GAA provides a superior correction of the skeletal muscle

phenotype, despite the high peak enzyme activity levels observed with 20 mg/kg rhGAA infusions (Fig. 1C).”

Fig 2A is confusing because the scales on the y axis are very different for each panel. I would suggest it needs an extra panel to indicate the enormous differences in the level of enzyme seen in different muscles. I would suggest comparing highest dose expression across the different tissues. These disparities then need to be picked up in the text and discussed.

ANSWER: We agree with the Reviewer that having an additional figure comparing GAA activity across tissues would be beneficial. The panel B of the edited Figure 2 now shows the comparison of the levels measured in the different tissues. We thank the Reviewer for the suggestion, as the comparison better shows full alignment of our data with previous reports. We added a comment on the new panel 2B in the Results section of the manuscript:

“The comparison of GAA activity across muscle groups confirmed the more efficient enzyme uptake in heart and diaphragm compared to quadriceps and triceps (Fig. 2B).”

I have some problems with the sections on enzyme delivery to the CNS. Firstly CNS disease is not a significant feature in the natural history of Pompe disease; it has recently come to light as a late complication in patients with IOPD who have been successfully treated with ERT, which is in fact lifesaving in this condition. To suggest that delivery to the CNS is a major unmet clinical need is inaccurate.

ANSWER: We understand the Reviewer’s concern. We surveyed the literature and discussed extensively the issue of CNS involvement in Pompe disease with several clinicians and scientists. Several preclinical studies by us and others reported CNS glycogen storage accompanied with a neurological phenotype^{2, 3, 10, 11, 12, 13, 14}, then, worsening of CNS inflammation has been shown to enhance the respiratory phenotype of Pompe mice⁴. In the clinic, CNS alterations have been documented in infantile Pompe patients^{15, 16, 17, 18, 19, 20} and alterations of brain vasculature^{21, 22} and CNS have been observed in adult patients²³. Loss of functional motor neuron outputs, documented in IOPD patients, also seems to represent a concern for proper diaphragm activation²⁴. The outcome of our survey is that CNS alterations are an important concern in the clinical management of Pompe disease, thus we focused our efforts in evaluating the effect of gene therapy on this organ in addition to muscle.

We addressed the Reviewer’s concern we modified the language in the manuscript to reflect that CNS features of PD are emerging and perhaps not fully established:

Introduction section, we added few references on the CNS manifestations of Pompe disease.

Results section:

“Glycogen accumulation in the central nervous system (CNS) **contributes is emerging as a feature contributing** to PD pathology in humans.”

Discussion section:

“Moreover , additional data coming from long-term follow up of PD patients will strengthen the link between glycogen accumulation in lysosomes and CNS pathology.”

Secondly, there is no satisfactory explanation of how enzyme is accessing the CNS. It is widely accepted that lysosomal enzymes in the blood do not cross the blood brain barrier. The data presented here shown that enzyme can enter endothelial cells, which has already been well demonstrated in other LSDs, but not that it can be exported again into the brain. This data is a misleading distraction and should be omitted. Instead, they should consider other explanations: either that enough vector genomes enter the CNS to lead to significant gene expression (there does not appear to be any data here about where vector genomes are found in mice, which would be useful, at least in the supplementary material); or that expression is from bone marrow derived cells (where they have found significant vector genomes in NHP) repopulate g the CNS, which is a well recognised mechanism.

ANSWER: We understand the Reviewer concern. We and others have shown that the proteins expressed in the liver and present at high levels in the circulation cross the blood-brain barrier ^{2, 25}. This is also well described in the context of monoclonal antibodies, which can access the CNS, in small quantities, when delivered as an intravenous bolus ²⁶. In the context of Pompe gene therapy with secretable GAA expressed in the liver, we previously showed that transgene expression driven by the liver-specific promoter hAAT in CNS is not significant ²⁷. Of note, AAV8 vectors do not significantly cross the blood-brain barrier and also do not transduce efficiently bone marrow cells. Our postulated hypothesis is that GAA is taken up by endothelial cells and transferred across in a transcytosis process. This has been described in the context of engineered AAV vectors targeting the brain endothelium ²⁸ and recent work in the context of modified lysosomal proteins showed that transcytosis can be exploited to improve brain targeting of drugs ²⁹. Nevertheless, we removed the endothelial cell uptake experiment to address the Reviewer’s concern on the preliminary nature of the findings.

To clarify that different mechanisms could be involved in GAA transport across the blood-brain barrier, we added the following sentence to the Discussion section:

“**Leakage across the BBB ²⁵, transport via exosomes ³⁰, or lysosomal exocytosis at neuromuscular junctions ³¹, could contribute to the transport of GAA from the systemic circulation to the CNS.**”

I am not sure what the added value of the chaperone work is. If the liver directed gene therapy approach works so well, then why would we need a further, expensive, pharmacological intervention. They have no evidence that this would provide added clinical benefits. Without that I don't think it adds anything to the manuscript.

ANSWER: We understand the Reviewer's concern. As Reviewer 2 stated, we believe that the chaperone work presented here is of interest for the readership for a number of reasons. The use of chaperones is being explored for a number of diseases, including Pompe, as a way to increase the efficacy of ERT^{32, 33} or as a monotherapy³⁴. This means that, potentially, the combination of gene therapy with chaperones will be a possible scenario. Of note, additive therapies have been used in combination with gene therapy in the context of spinal muscular atrophy³⁵, in which children treated with an AAV9 expressing the SMN transgene at subtherapeutic levels received also the ASO Spinraza. While the ASO treatment is not a chaperone, it provides an example of combination therapy. Our data show that chaperone administration does result in enhanced efficacy, thus provides a potential avenue to 1) reduce the therapeutic dose of vector by developing a combination therapy 2) provide an add on solution in case of subtherapeutic levels of expression or partial loss of expression of the transgene over time.

We clarified these points in the Discussion section of our manuscript:

“While the need to supplement gene therapy with PCs will have to be evaluated in the clinic, the combination of gene therapy with pharmacological approaches to enhance efficacy is an important concept that has previously been explored, for example, **in mice in the context of autoimmunity and in clinical trials of gene therapy for spinal muscular atrophy³⁵. In PD gene therapy, the use of PCs can provide a potential avenue to reduce the therapeutic dose of vector or to boost enzyme activity in tissues in case of subtherapeutic levels of expression of the GAA transgene.**”

In the Discussion, the comparison with haemophilia (line 432) is not really appropriate as factor VIII is a protein which is normally secreted from the liver and acts in plasma rather than having to be taken up to cells: delivering the factor VIII gene to the liver in haemophilia is actually correcting the affected tissue, which is not what is being proposed here at all.

ANSWER: The Reviewer is correct, factor IX is naturally produced by hepatocytes while factor VIII is made in endothelial cells. Both proteins act directly in blood. We believe the hemophilia example is relevant as it shows that the expected therapeutic range extrapolated in the context of protein replacement therapy (12% and above,³⁶) does not apply to gene therapy, in which even levels of ~5% of normal show clear therapeutic benefit (e.g.³⁷). The Reviewer also highlight an important feature of our work, which is in fact that the enhanced therapeutic benefit is manifested not only in blood but also in a context in which enzyme uptake from a distal tissue is required.

We amended the section of the Discussion describing the hemophilia findings as an example of the potential of gene therapy, to clarify these points and specifically explain why we believe citing the experience from hemophilia trials is relevant in the context of the current work.

“An example of the favorable pharmacokinetics offered by gene therapy comes as ~~they support findings across several~~ from liver ~~gene therapy~~ trials for hemophilia, showing an almost complete correction of the bleeding diathesis in patients over a

broad range of clotting factor levels, even lower than those predicted based on protein replacement therapy³⁶. ~~Importantly,~~ While the context of hemophilia differs from that of Pompe disease, as clotting factors act directly in blood and not in peripheral tissues, human trial data show that steady state levels of transgene expression positively affect clinical outcomes. Thus, the superior pharmacokinetics of liver-expressed protein therapeutics, demonstrated here, can potentially offer opportunities to develop more efficacious therapeutic interventions across a broad range of indications.”

The sentence about CNS glycogen storage having an important role in respiratory failure in Pompe (starting on line 450) is not evidence based and should be omitted. This is part of the authors unjustified attempt to portray CNS delivery of enzyme as an unmet need. Their data on muscle delivery is good enough, and they really don't need to make doubtful claims about CNS delivery.

ANSWER: We believe this concern was already addressed as response to a previous comment.

Reviewer #2 (Remarks to the Author):

The manuscript NCOMMS-21-06438, entitled "Hepatic expression of GAA results in enhanced enzyme bioavailability in mice and non-human primates" by Mingozzi and colleagues, reports on hepatic gene transfer with adeno-associated virus vectors expressing secretable GAA. It is a very thorough study that is well described and elegantly conducted with state-of-the-art methods. The investigation with a Pompe disease mouse model reveals great therapeutic potential for hepatic gene transfer with AAV expressing secretable GAA. This finding should raise interest in the field of lysosomal storage diseases. The strength of the work is the careful comparison of the corrections brought about by the AAV intervention with those reached with enzyme replacement therapy (the present therapy for Pompe disease). Of interest is also the reported examination of the potential additive value of administration of small compound chaperones to AAV treated Pompe mice.

ANSWER: We would like to thank the Reviewer for appreciating the relevance of our work, particularly the importance of providing careful side-by-side comparison of the potential therapeutic efficacy of gene transfer (with or without chaperone) vs. conventional protein therapy.

Specific comment to authors.

The study is well conducted with attention to detail.

One wonders whether it is not worthwhile to also consider an analysis of

unrelated lysosomal enzyme levels in tissues of treated and untreated PD mice. Often in LSDs, other lysosomal enzymes are secondary elevated in response to lysosomal storage. A successful clearance of excessive glycogen by the AAV intervention would likely be accompanied by corrections in other lysosomal enzymes as well. Demonstration of such corrections would further demonstrate efficacy of the AAV intervention.

ANSWER: The Reviewer raises an important question. The alteration of multiple lysosomal enzymes has been previously demonstrated, for example in a mouse model of mucopolysaccharidosis type IIIC³⁸. It is indeed possible that accumulation of GAA and consequent lysosomal dysfunction would affect other enzymes. While this is an interesting question, we currently do not have capabilities to analyze other enzymes at the protein level, which we intend to do in future studies. However, based on the Reviewer's comment, we went back and analyzed muscle tissue transcriptomics data from a published study³ in which we treated 9-month-old Pompe mice with the same AAV vector used in the study. As shown in the figure below, we were able to observe changes in mRNA levels for several lysosomal enzymes in Pompe mice vs. wild type animals (PBS *Gaa*^{+/+} vs. PBS *Gaa*^{-/-} in the figure below) as described By Marco and colleagues³⁸. This was likely exacerbated by the old age of the mice in the study. In this setting, hepatic gene transfer with and AAV8 vector expressing secretable GAA (secGAA *Gaa*^{-/-}), identical to the one used in the current study, resulted in a variable extent rescue of the alterations, consistent with the beneficial effect of secGAA supplementation and, again, with the advanced disease state of these animals, which may have affected the ability to completely reverse the disease phenotype. Of note, in the current study we showed normalization of lysosomal function markers in AAV vector-treated mice, which may have a positive effect on the levels of lysosomal enzymes.

We added a sentence to the Discussion section of the manuscript to reflect these considerations:

“Similarly, the effect of secretable GAA on lysosomal function normalization, and potentially on other lysosomal enzymes³⁸, remain to be assessed.”

Because the transcriptomic data comes from a different study, our preference would be not to include the figure below in the current manuscript.

Figure legend. Heatmap representing relative expression values for lysosomal enzymes in *GAA*^{+/+}, *GAA*^{-/-} and secGAA-*GAA*^{-/-} samples from a previously published study, in which the Pompe phenotype of 9 month-old *GAA*^{-/-} mice was

rescued by AAV-secGAA treatment. The list of lysosomal enzymes was taken from Marcó et al (2016). Normalized expression values for GAA+/+, GAA-/- and secGAA-GAA-/- samples were recovered from GEO entry GSE150935³, averaged by condition and standardized for each gene. IDS, iduronate-2-sulfatase; HEXA, β -hexosaminidase A; GUSB, β -glucuronidase; GALNS, N-acetylgalactosamine-6-sulfatase.

References

1. Ferla R, *et al.* Non-clinical Safety and Efficacy of an AAV2/8 Vector Administered Intravenously for Treatment of Mucopolysaccharidosis Type VI. *Mol Ther Methods Clin Dev* **6**, 143-158 (2017).
2. Puzzo F, *et al.* Rescue of Pompe disease in mice by AAV-mediated liver delivery of secretable acid alpha-glucosidase. *Sci Transl Med* **9**, (2017).
3. Cagin U, *et al.* Rescue of Advanced Pompe Disease in Mice with Hepatic Expression of Secretable Acid alpha-Glucosidase. *Molecular therapy : the journal of the American Society of Gene Therapy*, (2020).
4. Colella P, *et al.* Gene therapy with secreted acid alpha-glucosidase rescues Pompe disease in a novel mouse model with early-onset spinal cord and respiratory defects. *EBioMedicine* **61**, 103052 (2020).
5. Han SO, *et al.* Low-Dose Liver-Targeted Gene Therapy for Pompe Disease Enhances Therapeutic Efficacy of ERT via Immune Tolerance Induction. *Mol Ther Methods Clin Dev* **4**, 126-136 (2017).
6. Doerfler PA, *et al.* Copackaged AAV9 Vectors Promote Simultaneous Immune Tolerance and Phenotypic Correction of Pompe Disease. *Hum Gene Ther* **27**, 43-59 (2016).
7. Rahemtulla A, *et al.* Normal development and function of CD8+ cells but markedly decreased helper cell activity in mice lacking CD4. *Nature* **353**, 180-184 (1991).
8. Joseph A, Munroe K, Housman M, Garman R, Richards S. Immune tolerance induction to enzyme-replacement therapy by co-administration of short-term, low-dose methotrexate in a murine Pompe disease model. *Clin Exp Immunol* **152**, 138-146 (2008).
9. Masat E, *et al.* Long-term exposure to Myozyme results in a decrease of anti-drug antibodies in late-onset Pompe disease patients. *Sci Rep* **6**, 36182 (2016).
10. Clarke J, Kayatekin C, Viel C, Shihabuddin L, Sardi SP. Murine Models of Lysosomal Storage Diseases Exhibit Differences in Brain Protein Aggregation and Neuroinflammation. *Biomedicines* **9**, (2021).

11. Lee NC, *et al.* Ultrastructural and diffusion tensor imaging studies reveal axon abnormalities in Pompe disease mice. *Sci Rep* **10**, 20239 (2020).
12. Lee KZ, *et al.* Hypoglossal neuropathology and respiratory activity in pompe mice. *Front Physiol* **2**, 31 (2011).
13. DeRuisseau LR, *et al.* Neural deficits contribute to respiratory insufficiency in Pompe disease. *Proceedings of the National Academy of Sciences of the United States of America* **106**, 9419-9424 (2009).
14. Sidman RL, *et al.* Temporal neuropathologic and behavioral phenotype of 6neo/6neo Pompe disease mice. *J Neuropathol Exp Neurol* **67**, 803-818 (2008).
15. Byrne BJ, *et al.* Pompe disease gene therapy: neural manifestations require consideration of CNS directed therapy. *Ann Transl Med* **7**, 290 (2019).
16. Korlimarla A, Lim JA, Kishnani PS, Sun B. An emerging phenotype of central nervous system involvement in Pompe disease: from bench to bedside and beyond. *Ann Transl Med* **7**, 289 (2019).
17. Ebbink BJ, *et al.* Classic infantile Pompe patients approaching adulthood: a cohort study on consequences for the brain. *Dev Med Child Neurol* **60**, 579-586 (2018).
18. Kishnani PS, Beckemeyer AA, Mendelsohn NJ. The new era of Pompe disease: advances in the detection, understanding of the phenotypic spectrum, pathophysiology, and management. *Am J Med Genet C Semin Med Genet* **160C**, 1-7 (2012).
19. Spiridigliozzi GA, Heller JH, Kishnani PS. Cognitive and adaptive functioning of children with infantile Pompe disease treated with enzyme replacement therapy: long-term follow-up. *Am J Med Genet C Semin Med Genet* **160C**, 22-29 (2012).
20. Rohrbach M, *et al.* CRIM-negative infantile Pompe disease: 42-month treatment outcome. *J Inherit Metab Dis* **33**, 751-757 (2010).
21. Sacconi S, Bocquet JD, Chanalet S, Tanant V, Salviati L, Desnuelle C. Abnormalities of cerebral arteries are frequent in patients with late-onset Pompe disease. *J Neurol* **257**, 1730-1733 (2010).
22. Mormina E, *et al.* Intracranial aneurysm management in patients with late-onset Pompe disease (LOPD). *Neurol Sci* **42**, 2411-2419 (2021).
23. Musumeci O, *et al.* Central nervous system involvement in late-onset Pompe disease: clues from neuroimaging and neuropsychological analysis. *Eur J Neurol* **26**, 442-e435 (2019).

24. Smith BK, Corti M, Martin AD, Fuller DD, Byrne BJ. Altered activation of the diaphragm in late-onset Pompe disease. *Respir Physiol Neurobiol* **222**, 11-15 (2016).
25. Ruzo A, *et al.* Liver production of sulfamidase reverses peripheral and ameliorates CNS pathology in mucopolysaccharidosis IIIA mice. *Molecular therapy : the journal of the American Society of Gene Therapy* **20**, 254-266 (2012).
26. Sevigny J, *et al.* The antibody aducanumab reduces Abeta plaques in Alzheimer's disease. *Nature* **537**, 50-56 (2016).
27. Colella P, *et al.* AAV Gene Transfer with Tandem Promoter Design Prevents Anti-transgene Immunity and Provides Persistent Efficacy in Neonate Pompe Mice. *Mol Ther Methods Clin Dev* **12**, 85-101 (2019).
28. Chen YH, Chang M, Davidson BL. Molecular signatures of disease brain endothelia provide new sites for CNS-directed enzyme therapy. *Nat Med* **15**, 1215-1218 (2009).
29. Kariolis MS, *et al.* Brain delivery of therapeutic proteins using an Fc fragment blood-brain barrier transport vehicle in mice and monkeys. *Sci Transl Med* **12**, (2020).
30. Gonzales PA, *et al.* Large-scale proteomics and phosphoproteomics of urinary exosomes. *J Am Soc Nephrol* **20**, 363-379 (2009).
31. Andrews NW. Regulated secretion of conventional lysosomes. *Trends Cell Biol* **10**, 316-321 (2000).
32. Kishnani P, *et al.* Duvoglustat HCl Increases Systemic and Tissue Exposure of Active Acid alpha-Glucosidase in Pompe Patients Co-administered with Alglucosidase alpha. *Molecular therapy : the journal of the American Society of Gene Therapy* **25**, 1199-1208 (2017).
33. Koeberl DD, *et al.* Improved muscle function in a phase I/II clinical trial of albuterol in Pompe disease. *Mol Genet Metab* **129**, 67-72 (2020).
34. Lenders M, *et al.* Treatment of fabry disease with migalastat-outcome from a prospective 24 months observational multicenter study (FAMOUS). *Eur Heart J Cardiovasc Pharmacother*, (2021).
35. Mendell JR, *et al.* Five-Year Extension Results of the Phase 1 START Trial of Onasemnogene Apeparvovec in Spinal Muscular Atrophy. *JAMA Neurol*, (2021).
36. den Uijl IE, Fischer K, Van Der Bom JG, Grobbee DE, Rosendaal FR, Plug I. Analysis of low frequency bleeding data: the association of joint bleeds according to baseline FVIII activity levels. *Haemophilia* **17**, 41-44 (2011).

37. Nathwani AC, *et al.* Long-term safety and efficacy of factor IX gene therapy in hemophilia B. *The New England journal of medicine* **371**, 1994-2004 (2014).
38. Marco S, *et al.* Progressive neurologic and somatic disease in a novel mouse model of human mucopolysaccharidosis type IIIC. *Dis Model Mech* **9**, 999-1013 (2016).

Reviewers' Comments:

Reviewer #1:

Remarks to the Author:

Thank you for addressing my concerns.

Reviewer #2:

Remarks to the Author:

The authors replied correctly to the suggestion made for improving the manuscript. They added a relevant sentence in the Discussion. Although no further data were added, the answer, and motivation, by the authors is satisfactory.

Answers to Reviewers:

Reviewer #1 (Remarks to the Author):

Thank you for addressing my concerns.

ANSWER: We are extremely grateful to the reviewer for the help in improving the quality of our work.

Reviewer #2 (Remarks to the Author):

The authors replied correctly to the suggestion made for improving the manuscript.

They added a relevant sentence in the Discussion. Although no further data were added, the answer, and motivation, by the authors is satisfactory.

ANSWER: We are extremely grateful to the reviewer for the help in improving the quality of our work.

We hope we successfully addressed all editorial requirements.

Please let us know if you have any additional questions.

Kind regards,

Federico